# POMDP Planning for Object Search in Partially Unknown Environment

**Yongbo Chen**[*]
School of Computing,
Australian National University
Canberra, ACT, 2601, Australia
Yongbo.Chen@anu.edu.au

**Hanna Kurniawati**
School of Computing,
Australian National University
Canberra, ACT, 2601, Australia
Hanna.Kurniawati@anu.edu.au

## Abstract

Efficiently searching for target objects in complex environments that contain various types of furniture, such as shelves, tables, and beds, is crucial for mobile robots, but it poses significant challenges due to various factors such as localization errors, limited field of view, and visual occlusion. To address this problem, we propose a Partially Observable Markov Decision Process (POMDP) formulation with a growing state space for object search in a 3D region. We solve this POMDP by carefully designing a perception module and developing a planning algorithm, called Growing Partially Observable Monte-Carlo Planning (GPOMCP), based on online Monte-Carlo tree search and belief tree reuse with a novel upper confidence bound. We have demonstrated that belief tree reuse is reasonable and achieves good performance when the belief differences are limited. Additionally, we introduce a guessed target object with an updating grid world to guide the search in the information-less and reward-less cases, like the absence of any detected objects. We tested our approach using Gazebo simulations on four scenarios of target finding in a realistic indoor living environment with the Fetch robot simulator. Compared to the baseline approaches, which are based on POMCP, our results indicate that our approach enables the robot to find the target object with a higher success rate faster while using the same computational requirements.

## 1 Introduction

Finding an object in a partially known and cluttered environment is critical for many robotics tasks. For instance, a robot operating in human environments, such as our homes may need to find mugs, glasses, or books that may be placed on top of various furniture like beds, tables, or chairs and partially occluded by other objects. While the positions of the furniture are commonly known, the positions of the smaller objects, such as the mugs and glasses, are not generally known and move very often. Humans can typically find those smaller objects by relying on prior knowledge, assuming the object they are searching for has been seen before[1] and removing occlusions as necessary. This kind of target objects-oriented problems is also called "Mechanical Search" problem [7].

This paper proposes a planning method based on a novel solver, called Growing Partially Observable Monte-Carlo Planning (GPOMCP), that enables mobile manipulators to do just that: Finding a target object that might be occluded by other objects in a partially unknown environment by moving to positions with better visibility or by removing occlusion when necessary.

---

[*]Corresponding author

37th Conference on Neural Information Processing Systems (NeurIPS 2023).

We assume the robot is operating in a home environment that consists of multiple rooms and is provided with 3D point cloud and 2D occupancy grid maps of the environment and a set of the target object's photos. These maps contain the furniture but do not contain information about the objects on top of the furniture. GPOMCP computes an efficient search policy that utilizes onboard sensors and its ability to remove objects within a designated search region to find the target object even if it is partially or completely occluded while accounting for localization and perception errors. To handle the partial observability nature of the problem, GPOMCP is based on the Partially Observable Markov Decision Process (POMDP). A unique feature of GPOMCP is it can efficiently modify policies when the number of state variables increases. This feature allows GPOMCP to model the environment incrementally, starting from an environment map with no objects on top of the furniture to a more faithful map where all the objects on top of the furniture have been included in the map and everything in-between, *as needed* until the target object is found.

In addition to the above solver, we propose a POMDP model for the object-finding problem that exploits GPOMCP's ability to modify policies as the number of state variables increases. To this end, the model maintains a set of state variables that represents information about the robot's guess of the target object, which we refer to as a guessed target object. A belief over this set of state variables represents the robot's uncertainty over the characteristics of the target object, such as the pose, whether the object can be moved or not, etc. As new observations are perceived, the robot refines its belief about the guessed target object, and eventually, the belief has a high chance to converge to the correct target object positions based on suitable parameter settings. Experimental results on four different scenarios using a Fetch robot simulator in Gazebo indicate that the above model and GPOMCP significantly outperforms POMCP[20].

## 2  Background

A POMDP is a general mathematical formulation for a sequential decision-making problem where the state is not fully observable by the agent. Formally, it is defined as an 8-tuple $< \mathcal{S}, \mathcal{A}, O, T, Z, R, b_0, \gamma >$, where the state space $\mathcal{S}$ denotes the set of all possible states of the robot and the environment; the action space $\mathcal{A}$ denotes the set of all actions the robot can perform; the observation space $O$ means the set of all observations the robot can perceive. At each time step, a POMDP agent is at a state $s \in \mathcal{S}$, but this state is generally not known exactly by the agent and therefore the POMDP agent uses a belief $b$ —a distribution over the state space— to represent this state uncertainty. The set of all beliefs is referred to as the belief space and denoted as $\mathcal{B}$. At each time step, the agent performs an action $a \in \mathcal{A}$, which may result in the agent arriving at a new state. However, the effect of actions is uncertain and is represented by a conditional probability function, namely the transition function $T(s, a, s') = Pr(s'|s, a)$. Once the agent arrives at a new state (which may not be fully known), the agent perceives perceives an observation $o \in O$, governed by the observation function $Z(s', a, o) = Pr(o|s', a)$, which is a conditional probability function that represents the observation the robot may perceive after performing action $a \in \mathcal{A}$ in state $s' \in \mathcal{S}$. Together with perceiving an observation, the agent also receives an immediate reward, governed by the reward function $R(s, a, s') \in \mathbb{R}$, which can be parameterized by a state, an action, a state–action pair, or a tuple of state, action, and subsequent state. The estimate on the initial state is called the initial belief and denoted as $b_0$. Last but not least, $\gamma$ is a discount factor following $0 < \gamma < 1$ [13]. The goal of solving a POMDP problem is to find an optimal policy $\Pi^*(b) = \mathrm{argmax}_{a \in \mathcal{A}} Q(b, a)$ for belief $b \in \mathcal{B}$, where the $Q(b, a)$-value is the value of executing action $a$ when the agent is at belief $b$ and continuing optimally afterwards.

POMDP solvers can be broadly classified into two main classes based on their real-time performance: offline and online solvers. While offline solvers can achieve optimal solutions for small-scale problems, such as the point-based POMDP algorithm [19, 14], they require too much memory for real-world applications. Most online POMDP solvers commonly explore the sparse belief sampling with a collection of particles and perform a Monte-Carlo tree search (MCTS) to grow the belief tree within the limited planning time. State-of-the-art online methods include Partially Observable Monte Carlo Planning (POMCP) [20], Adaptive Belief

Tree (ABT) [15], Determinized Sparse Partially Observable Tree (DESPOT) [21], Partially Observable Monte Carlo Planning with Observation Widening (POMCPOW) [22], and Adaptive Discretization using Voronoi Trees (ADVT) [9] methods. Among them, POMCP and POMCPOW methods rely on the Monte Carlo backup, while others use the Bellman backup. Although most of these methods are designed to explore belief nodes more deeply in continuous or large action and observation spaces, continuous state, action, and observation spaces continue to pose a significant challenge.

A substantial category of mechanical search methodologies is closely intertwined with advancements in learning technologies, including deep reinforcement learning (RL) [12], deep Q-learning [25], receding horizon planning guided by neural network-based distribution [2], deep-geometric inference systems [7, 10]. For example, in [12], the authors use a deep RL procedure that combines teacher-aided exploration, a critic with privileged information, and mid-level representations to uncover a target object occluded by a heap of unknown objects. Except for the learning-based methods, some other methods exploit the classical technologies, like the adaptive horizon exploration algorithm [8] and mixed-integer program (MIP) [3]. A shared characteristic among these methods is their specialized application in specific environments, primarily centered around scenarios involving shelves and boxes.

Some other works on object search are solved using POMDP planning approaches, such as a recent MCTS-based mechanical search method that relies on a 1D occupancy distribution for the current camera observation in cluttered shelves. [11]. The Object-Oriented POMDP (OO-POMDP) [24], which is very popular, considers the state and observation spaces to be factored by a set of $n$ objects, where each belongs to a class with a set of attributes, and the beliefs about different objects are independent for the small computational reason. Many previous works also limit the workspace to a 2D plane with simple transition and observation models. For example, Aydemir et al. proposed a 2D POMDP formulation for the object search in a room by calculating candidate viewpoints [1]; The object search task has been extended to 3D with an object size-friendly property using a multi-resolution planning algorithm [28]. We follow the same Object-Oriented idea because of the good applicability of the real-world application but go forward one step to consider the occlusion relationship for the belief update between different objects in a complicated 3D environment, including some challenging cases with partial and fully occluded objects.

## 3 POMDP formulation for object search

Our focused object search task is to find a movable target object $\{obj_{tar}\}$ in several clustered workspaces that contain an unknown number $n > 1$ of obstacle objects $\{obj_{obs}\}$ with unknown locations. To facilitate robot navigation, we have access to point cloud and occupancy grid maps representing the planning environment, including furniture. However, we do not possess the point clouds of the obstacle objects $\{obj_{obs}\}$ or the target object $obj_{tar}$. Therefore, during the execution process, we rely on on-board sensors to detect and identify these objects $(obj_{obs} \cup obj_{tar})$. Since we gather information about the objects using real-world sensors, they are not mutually independent in our scenario and may influence each other. This differs from the assumption made in OO-POMDP.

**State space $\mathcal{S}$.** A state $s = \{s_r, s_{o^0}, s_{o^1}, \cdots, s_{o^n}\} \in \mathcal{S}$ is represented as two parts including robot configuration and object-oriented factors, where $s_r$ is the robot state and $s_{o^i}$ denote the object state of the $i$-th object. A robot state is written as $s_r = (p, l_h, l_p, l_t)$, where $p$ is the 6D robot base pose, $l_h$ is the lift height, $l_p$ and $l_t$ are the pan and tilt angles of the robot head. At the beginning stage, the robot may not get into the workspace, so no detected object is available. In order to guide the robot toward meaningful actions, we introduce a guessed target object $s_{o^0} = (p_{o^0}, s^x_{o^0}, s^y_{o^0}, s^z_{o^0}, g_{o^0}, m_{o^0}, u_{o^0}) \in \mathbb{R}^{1\times20}$, where $p_{o^0} = (p^p_{o^0}, p^o_{o^0})$ is the 6D object pose (3D position $p^p_{o^0}$ and 3D rotations represented by quaternion $p^o_{o^0}$); $s^x_{o^0}$, $s^y_{o^0}$, and $s^z_{o^0}$ mean the sizes along the principal 3D axis of the object; $g_{o^0} \in \mathbb{R}^{1\times8}$ denotes 8 grid odds values to identify whether this object is a target or obstacle object, which will be updated using the scores obtained from image matching. $m_{o^0} \in \mathbb{R}$ is the odds value to evaluate the move-ability of the guessed target object; $u_{o^0} \in \{-2, -1, 0, 1\}$ is the int front value to claim whether this object is updating or not. Specifically, $-2$ means this object is

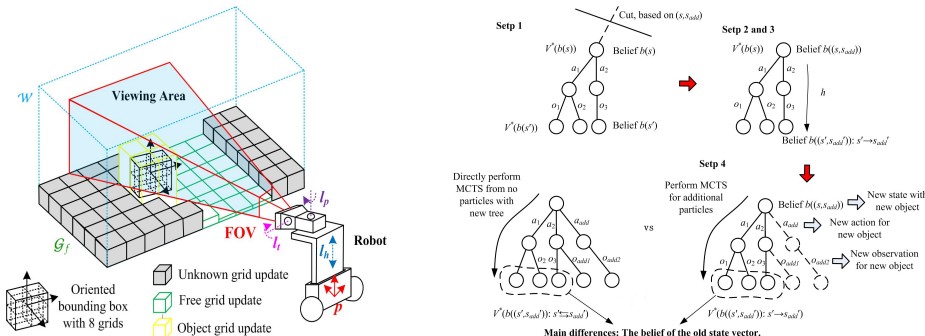

Figure 1: Grid update in one frame          Figure 2: Main steps for GPOMCP

removed after performing the removing action; -1 denotes that the object is still updating without performing declaring and removing actions, 0 and 1 means this object has been declared as an obstacle object and a target object. The belief of the position $p_{o^0}^p$ of the guessed target object is saved in a grid world $\mathcal{G}_f$ generated by all workspaces $\mathcal{W}$ using many odds values $Odd(\mathcal{G}_f)$ with a given resolution. Inside each grid, the position is sampled uniformly and its probability is obtained by its odds value. The odds values are updated based on the field of view (FOV) of the camera using real-world measurement in the excursion process, which is similar to the update of the occupancy grid map in Simultaneous Localization and Mapping (SLAM) [4], following:

$$\log Odd(\mathcal{G}_f|z_{1:t}) = \log Odd(\mathcal{G}_f|z_t) + \log Odd(\mathcal{G}_f|z_{1:t-1})$$
$$Odd(\mathcal{G}_f|z_{1:t}) = P(\mathcal{G}_f|z_{1:t})/P(\neg\mathcal{G}_f|z_{1:t}) \tag{1}$$

where $P(\mathcal{G}_f|z_{1:t})$ and $P(\neg\mathcal{G}_f|z_{1:t})$ means the probability of the object belonging to and not belonging to this grid based on multiple observations $z_{1:t}$; $Odd(\mathcal{G}_f|z_{1:t})$ is the corresponding odd value. In the belief tree search, we do not update the odds values $Odd(\mathcal{G}_f)$ and it will be updated after the real-world excursion. In the planning stage, this grid world is just used to sample the potential position of the guessed target object in the root node. The guessed target object is special with a constant orientation (set as (0, 0, 0, 1)), size (0.1, 0.1, 0.1), and move-ability value (set as 100, movable). The grid values $g_{o^0}$ and the declared value $u_{o^0}$ are update-able in the belief tree search but need to be reinitialized as the given value after each excursion. Fig.1 shows the scenario about the grid world for the guessed target object. In fact, the guessed target object variable within the state vector saves the belief about the position of the target object. It not only encodes the probability information derived from the grid world with odds values but also facilitates efficient planning within Monte Carlo tree search with low computational complexity, acceptable memory cost, and coding friendliness. More explanations to the guessed target object are shown in the supplementary materials. The other object $s_{o^i}$, $i \neq 0 \in \mathbb{R}^{1\times20}$ follows the same format and all parameters should be updated both in belief tree search and real-world excursion. It is noted that the pose of the other objects $s_{o^i}$ is estimated based on the point cloud which is independent of the grid world $\mathcal{G}_f$. The newly detected objects after each real-world excursion will be appended to the state vector and form a growing state space $(s, s_{add}) \in \mathcal{S} \times \mathcal{S}_{add}$.

**Action space $\mathcal{A}$.** Our method is task-level planning with many primitive actions instead of control-level planning. The key purpose of setting these actions is to change the FOV of the robot camera to cover the workspace and further operate objects. The action space consists of 5 types of primitives: $Move(p^t, \omega_m)$ means to move the robot base to a given target pose $p^t$; $Head(l_p, l_t)$ denotes the actions to change the angles of the robot head pan (left/right) and tilt (up/down) to $l_p$ for the pan and $l_t$ for the tilt; $Lift(l_h)$ represents to move the robot height to $l_h$; $Declaring(u_{o^i})$ means to declare the updating object ($u_{o^i} = -1$) to be the target object ($u_{o^i} = 1$) or the obstacle object ($u_{o^i} = 0$); $Removing(i, \omega_r)$ is the action to remove $i$-th object using robot manipulator. All actions are assumed to have a 100% success rate for simplicity[2].

---

[2]Even in the gazebo Fetch simulator, it is hard to ensure that each action is successfully performed, especially for removing actions. In order to guarantee the 100% success rate, after the robot's common control level operation, we call the gazebo server "set_model_state" to physically set the model state with some Gaussian noises, which will be improved in future work.

**Observation space** $O$**.** The main focused observation is a FOV viewing frustum $V$ projected from the robot head camera. Because we view each object by an oriented box with 8 grids, for each observation, we assume to update at most the 4-nearest grids with a noise negative log-odds value $-c_o + \eta$, a noise positive log-odds value $c_o + \eta$, a noise near-zero log-odds value $\eta$, where $\eta$ follows the Gaussian distribution. Following the blind spot caused by the other objects, sometimes, the robot camera can not detect all 4-nearest grids. We set the observation space to be discrete. By fusing all $n_o$ detected objects, we can get the observation to be $\{\{i, \cdots, j\}, \{o(i), \cdots, o(j)\}\}$, $|\{i, \cdots, j\}| = n_o$, where $\{i, \cdots, j\}$ records the observed object with at least one updated grid and $\{o(i), \cdots, o(j)\}$ are their updated log-odds value.

**Transition function** $T$**.** If the actions $Move(\boldsymbol{p}^t, \omega_m)$, $Head(l_p, l_t)$, and $Lift(l_h)$ are applied to the robot, the camera location and orientation will change with some noises. The camera motion follows the rigid transformation and the static structure information of the applied robot is read from $tf$ message. The declaring action $Declaring(u_{oi})$ will change the robot status $u_{oi}$ based on the comparison between the mean value of the minimal 2 log-odds of 8 grid values and the thresholds $C_d^o$ for the obstacle object and $C_t^o$ for the target object. If no removing action is performed to the $i$-th object, the $i$-th object is static and rigid. The removing action $Removing(i, \omega_r)$ on $i$-th object will move the $i$-th object outside the workspace and set the status $u_{oi}$ to -2, which means they will not block the observation of the other objects.

**Observation function** $Z$**.** The coordinates of 8 grids of the $i$-th object can be computed using center point $\boldsymbol{p}_{oi}^p$ and the orientation $\boldsymbol{p}_{oi}^o$ of the object pose and its size $s_{o^0}^x, s_{o^0}^y, s_{o^0}^z$ estimated by point cloud. After performing one action, the visible grids of different objects belonging to the 4 nearest grids will be updated. The updated value will be decided based on the mean log-odds value of all grids. If the original mean log-odds value is larger than a positive threshold $v_p > 0$, the updated value will be set to be positive $c_0 + \eta$. If it is smaller than a negative threshold $v_n < 0$, the updated value will be set to be negative $-c_0 + \eta$. Otherwise, the value between the positive and negative thresholds will generate a near-zero $\eta$. So as to encourage the exploration of the robot, the initial grid values $g_{o^0}$ of the guessed target object are reinitialized to be larger than the positive threshold $v_p > 0$ after each real-world excursion.

**Reward function** $R$**.** Our expected sequential actions involve the robot moving to the workspace for observation, continuously changing its viewpoints to gather more measurements, updating odds values in 8 grids, then declaring the updating object to be the obstacle or target object, and finally removing the obstacle objects for the more visible area or directly the target object to complete the whole task. Hence, if the whole task is completed successfully with the removal of the target object, the robot will receive a large positive reward $R_{max} \gg 0$. As the prerequisite action, correctly identifying the obstacle and target objects yields relatively large rewards ($R_{c_o}$ and $R_{c_t}$, respectively), where $R_{max} \gg R_{c_t} > R_{c_o} \gg 0$. All actions incur a negative step cost ($R_{min} < 0$). The cost of the removal action $Removing(i, \omega_r)$ is set as $2R_{min}$ due to its complexity. Illegal actions incur a large negative penalty ($R_{ill} \ll 0$), including collisions with occupied grids in the occupancy grid map, attempting to remove immovable objects, declaring an object that doesn't meet the condition, removing updating objects without declaration, and trying to remove objects that have already been removed.

## 4 Action execution and perception with on-board sensors

Our perception part is carefully explored with many useful outputs, like estimated object pose, estimated object size, object move-ability, and object detector, fusing both point cloud data and the image data without using manual quick response (QR) codes. Our framework is suitable for all mobile robots with 2D Lidar and RGBD cameras, but specifically, we mainly consider the Fetch robot here. Move base actions are implemented using a ROS interface move_base and the interaction with the AMCL-based navigation stack [18]. The robot lift height and head joints are controlled by following the joint-space trajectories on a group of joints based on a ROS interface ros_control [5]. Removing action is to pick up the object and place it in some given areas outside the workspace using the ROS tool moveit [6].

To simplify the scenario and reduce the navigation effort, we assume the availability of a pre-built point cloud map $\mathcal{M}$ of the robot's environment, including furniture and known

objects, which can be created beforehand through environment mapping and reused during planning. Then, the 3D point cloud $\mathcal{F}_j$ in $j$-th frame from the depth camera is matched with the map $\mathcal{M}$ by iterative closest point (ICP) [26]. The 3D ICP is initialized by matching 2D Lidar points with the 2D occupancy grid map. As the frame number increases, the point clouds of objects and environment $\mathcal{P}_j = \mathcal{M} \bigcup \mathcal{F}_0 \bigcup \cdots \bigcup \mathcal{F}_j$ are becoming more and more complete. After removing the original map $\mathcal{M}'_j = \mathcal{P}_j / \mathcal{M}$, the point clouds for $n$ newly detected objects $\{o_0, \cdots, o_{n-1}\} \in \mathcal{M}'_j$ are extracted by point cloud segmentation using the Euclidean cluster extraction method. For estimating object pose and size, we obtain the minimum oriented bounding box for each object using principal component analysis. The point cloud segmentation is applied followed by data association based on the Mahalanobis distance of centroid points and point-wise mean distance. Additional details can be found in the supplementary materials.

So as to complete the given object detection task using several given images and some semantic words (optional), based on the previous point cloud segmentation, we perform the point cloud re-projection, sub-image fusion, YOLO detection, and SIFT matching [17] to compute the probability of object detection. We also would like to manipulate the objects in the workspace to free some FOV, so we estimate the move-ability $r_{move}$ of each object by the learning-based Grasp Pose Detection (GPD) toolbox [23], the k-means clustering algorithm, and the ROS moveit toolbox. The details of the implementation are presented in supplementary materials. In the planning stage, for each particle, we will randomly sample a value for this object and compare it with the move-ability $r_{move}$ to identify its move-ability.

## 5 Belief tree reuse for online POMDP solver with growing state space

Here, we present a POMDP solver, GPOMCP, reusing the previous belief tree when the state space of the problem is growing. In this kind of POMDP problem, the observation obtained by real measurement causes the unpredicted new state, which means that the whole belief tree should be cut and a totally new belief tree should be computed by sampling. It is time-consuming. As shown in Fig. 2, we solve the problem by following steps:

- **Step 1**: When state space grows from $s \in \mathcal{S}$ to $(s, s_{add}) \in \mathcal{S} \times \mathcal{S}_{add}$, in the old belief tree $\mathcal{T}$, we can pick out the branches corresponding to old sub-state $s$ and set it as the root of new tree $\mathcal{T}'(s)$.
- **Step 2**: This new tree $\mathcal{T}'(s)$ has many particles with history $h$. For each particle, we can update the future state $s'$ by the added state vector $s'_{add}$ corresponding to new objects based on the transition function $T(s_{add}, a, s'_{add})$ and observation function $Z(s'_{add}, a, o_{add})$ [3]. It is noted that, in the state updating, we should consider the effect $s' \rightarrow s'_{add}$ from the old state vector $s$ corresponding to the old object. So we can get the updated histories about some or all particles within the given time limitation.
- **Step 3**: Then, starting from the root, we can grow the belief tree $\mathcal{T}'(s)$ using these updated particles [4]. It is noted that, in building the new belief tree $\mathcal{T}'(s)$, the new observation nodes with newly detected objects may be introduced, which means that the generated belief tree $\mathcal{T}'(s)$ may be not a sub-tree of the old belief tree $\mathcal{T}$.
- **Step 4**: Then, we perform the Monte Carlo tree search (MCTS) for all branches including the new actions and new observations. The action selection strategy follows the upper confidence bound (UCB) and the belief distance $C_b \sum_{o \in O} Z(s'_{new}, a, o') \| b((s', s'_{add}) : s' \rightarrow s'_{add}) - b((s', s'_{add}) : s' \rightleftarrows s'_{add}) \|_1 + V^*(b, a) + C_N \sqrt{\frac{logN(b)}{N(b,a)}}$, where $C_b$ and $C_N$ are coefficients set by users.

We know that the current optimal action should be selected following equation Eq. (2).

$$\max_{a \in A} \left( \sum_{s_{new} \in S} R(s_{new}, a, s'_{new}) b(s_{new}) + \gamma \sum_{o \in O} \sum_{s'_{new} \in S'} Z(s'_{new}, a, o') T(s_{new}, a, s'_{new}) V^*(b(s'_{new} : s' \rightleftarrows s'_{add})) \right) \quad (2)$$

---

[3] It is noted that because our considered actions $a_1$ and $a_2$ just perform on the old objects, the reward function does not need to be updated. Our reward function follows $R((s, s_{add}), a_1, (s', s'_{add})) = R(s, a_1, s')$, because the old action $a_1$ and $a_2$ just operate one object each step.

[4] Here, the belief of the old state **does not** consider the effect from the new object $s'_{add} \not\rightarrow s'$.

where $s_{new} = (s, \varnothing)$ and $s'_{new} = (s', s'_{add})$. Considering two ways (GPOMCP and the direct resampling way), in the sub-tree of the new belief tree, the only difference is from the last terms $V^*(b(s'_{new} : s' \rightleftarrows s'_{add}))$ (correct by building tree) and $V^*(b(s'_{new} : s' \rightarrow s'_{add}))$ (GPOMCP, without considering partial effect). Then, we have the following Theorems:

**Theorem 1.** *We are considering the unweighted particles for approximating the extended full-correct belief $b(s'_{new} : s' \rightleftarrows s'_{add}) = \frac{1}{N} \sum_{i=1}^{N} \mathbb{I}(s'_{new} = s^i_{new}{}' : s' \rightleftarrows s'_{add})$ and the reused approximated belief $b(s'_{new} : s' \rightarrow s'_{add}) = \frac{1}{N} \sum_{i=1}^{N} \mathbb{I}(s'_{new} = s^i_{new}{}' : s' \rightarrow s'_{add})$. Assuming the reward function is Borel measurable and bounded, denoted $R_{max} = \|R\|_\infty$, the belief L1 distance is limited in $\|b(s'_{new} : s' \rightleftarrows s'_{add}) - b(s'_{new} : s' \rightarrow s'_{add})\|_1 \le \delta$, and the optimal action selected by building the whole tree is same as the one selected by approximated value $\max_a V(b(s_{new}), a) = \max_a \hat{V}(b(s_{new}), a) = a^*$, the optimal value function of POMDP problem using our method $V^*(b(s_{new}))$ and the optimal value function using the direct resampling way $\hat{V}^*(b(s_{new}))$ will satisfy the following bounding equation:*

$$\|V^*(b(s_{new})) - \hat{V}^*(b(s_{new}))\|_1 \le \frac{\gamma R_{max}}{1 - \gamma} \delta \tag{3}$$

*Proof.* Please refer to the supplementary material. □

This theorem indicates that the reuse technology works well with a performance bound when the reused belief tree shows good approximations to the exact accurate belief tree. To enhance the estimation of the upper bound, we introduce a term $C_b \sum_{o \in O} Z(s'_{new}, a, o')\|b(s'_{new} : s' \rightarrow s'_{add}) - b(s'_{new} : s' \rightleftarrows s'_{add})\|_1$ to the UCB strategy when reusing the previous belief tree.

It is noted that Theorem 1 is correct only under the assumption $\max_a V(b(s_{new}), a) = \max_a \hat{V}(b(s_{new}), a) = a^*$, which means that there are some bounds for the performance guarantee shown in Theorem 1 using the reuse approximation.

**Discussion 1.** *In fact, our belief tree reuse can be considered as the wrong estimation $\triangle$ of the action and measurement node values in the correct belief tree. When the distance between the value function for the optimal action $V(b(s_{new}), a^*)$ and the value function for any sub-optimal action $V(b(s_{new}), a^*_{sub})$ is relatively large satisfying $V(b(s_{new}), a^*) - V(b(s_{new}), a^*_{sub}) >> \triangle$, which means that the estimation error $\triangle$ will not affect the final solution and the convergence of our solver follows the method directly rebuilding the whole tree.*

Following Discussion 1, we would like to get the exact performance bound to ensure that the obtained solution remains the same as the optimal solution using the fully-rebuilt belief tree. Firstly, we have the following corollary:

**Corollary 1.** *With the same assumption as Theorem 1, given the action $a$, the optimal action-value function of POMDP problem using our method $V^*(b(s_{new}), a)$ and the optimal value function using the direct resampling way $\hat{V}^*(b(s_{new}), a)$ will satisfy the following bounding equation:*

$$\|V^*(b(s_{new}), a) - \hat{V}^*(b(s_{new}), a)\|_1 \le \frac{\gamma R_{max}}{1 - \gamma} \delta \tag{4}$$

*Proof.* Just follows the similar proof process shown in Theorem 1, so we ignore it here. □

**Corollary 2.** *With the same assumption as Theorem 1, if the distance between the value function for the optimal action $V(b(s_{new}), a^*)$ and the value function for any sub-optimal action $V(b(s_{new}), a^*_{sub})$ is larger than $\frac{2\gamma R_{max}}{1-\gamma} \delta$, the optimal action obtained by the tree reuse approximation way will be same as the optimal action using the direct resampling way $\max_a V(b(s_{new}), a) = \max_a \hat{V}(b(s_{new}), a) = a^*$.*

*Proof.* Please refer to the supplementary material. □

The pseudocode Algorithms of the GPOMCP solver are shown in supplementary materials. Our proposed method follows the common procedure with four alternating stages, planning, execution, obtaining observation, and filtering. The optimal action for the current belief

is selected based on GPOMCP. Inside the POMDP solver, the Monte Carlo update [5] and a revised UCB strategy considering the belief error proposed in Section 5 are applied.

## 6   Simulations

In this section, we showcase our method's effectiveness via simulations conducted using C++ and Python codes within the Fetch robot simulator. The project operates on multiple Ubuntu 18.04 desktops equipped with an Intel Core i7-13700k processor, running exclusively on CPU. We evaluate our method using various object settings in different scenarios.

**Problem settings**   We implement our approach in a realistic gazebo living environment (Fig. 3a). The 3D point cloud map and 2D occupancy grid map, generated using [16], of this environment with furniture are available before the object searching task (Fig. 3b). There are multiple workspaces on the surfaces of the large furniture, like tables, beds, and desks, in this planning environment. Some unknown objects including a target object are set inside the different workspaces sometimes with an occlusion relationship. Four photos of the target object from different viewing directions are available for the object detector.

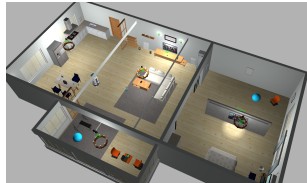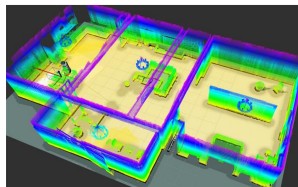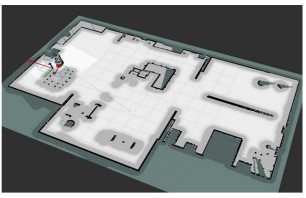

(a) Gazebo environment      (b) Corresponding point cloud and occupancy grid maps
Figure 3: The planning environment

We manually set several candidate poses that can observe partial workspace for the move base action $Move(\boldsymbol{p}^t, \omega_m)$. The candidate robot head motion $(l_p, l_t)$ are set within $l_p \in \{-\frac{1}{12}\pi, 0, \frac{1}{12}\pi\}$, $l_t \in \{-0.5, 0, 0.5\}$, and the lift motion $l_h$ is limited in $l_h \in \{0.0, 0.2, 0.4\}$. The FOV of the Fetch camera has a 60-degree horizontal view angle with a 480/600 height-width ratio. In the transition and observation function, using these actions, we add the nearest (0.5 meters to the camera center) and farthest planes (1.7 meters to the camera center) to the FOV and form a hexahedron. This hexahedron-shaped FOV will move based on the robot's motion and configuration change. As mentioned in Section 3, each object is represented as 8 grids based on its optimal oriented box, and the occlusion is computed using perspective projection of the cube vertices. The grid size of the grid world is set as 2 cm. A discussion about different grid sizes is shown in the supplementary materials. The log-odds value of these grids is initially set as unknown satisfying $\log \frac{0.5}{1-0.5} = 0$. These odds values will be updated after each real-world observation and then used to sample the potential position of the guessed target object. The planning time is limited to 60 s/step, the discounted factor $\gamma = 0.9$, the threshold to the object declaration $C_d^t = \log(8/2)$ and $C_d^o = \log(2/8)$, the thresholds for grid updating $v_p = 0.1$ and $v_n = -0.1$, the re-initialized grid values of the guessed target object are set as 0.2, the reward values $R_{max} = 10^5$, $R_{c_t} = 5 \times 10^4$, $R_{c_o} = 10^4$, $R_{min} = -1$, and $R_{ill} = -10^3$.

**Comparison results**   Because a lot of existing object search approaches are based on the pure POMCP method with different problem settings, eg. [27, 24], we compare our approach using Bellman (Ours_b) or Monte Carlo update (Ours_m) with the POMCP method without using/using the guessed target object (denoted as POMCP, POMCP_f). The comparison is implemented based on 4 different scenarios with different object numbers, including $LOOSE_1$ (4 objects in 1 workspace), $LOOSE_2$ (6 objects in 2 workspaces), $Hidden_1$ (7 objects

---

[5]Here, we use the Monte Carlo update because the visual measurement model in object search is not well predictable. The robot will frequently receive unexpected measurements, which are not in the belief tree. It breaks the advantage of the Bellman update (explore deeper) of Adaptive Belief Tree (ABT) solver [15]. We think that, in other applications, the Bellman update should be better.

in 1 workspace), and *Covered*$_1$ (7 objects in 1 workspace), as shown in Fig. 4. The target object is the blue snack box shown in the blue circle. Only the black headset box and the dish rack are not movable, and the others are all movable. Green arrows show the candidate robot poses. "Tick" means the target objects can be observed easily without changing the robot configuration. "Half tick" means the target can be observed if we change the robot configuration. "Cross" means the target object is fully hidden by the obstacle objects and cannot be seen from that pose even changing the robot configuration, including the lift of the robot, if the obstacle objects are not removed before. For each scenario, 20 trials are conducted and statistical results are reported. More resuls with different problem settings are shown in the supplementary materials.

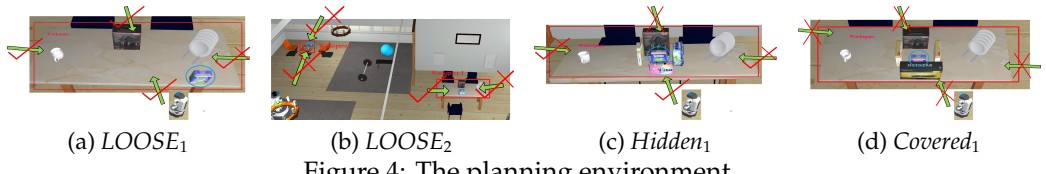

(a) *LOOSE*$_1$        (b) *LOOSE*$_2$        (c) *Hidden*$_1$        (d) *Covered*$_1$

Figure 4: The planning environment

The obtained 95% confidence interval of the discounted cumulative reward, the steps, and the success rate are reported in Table 1. For clarity, we present one sequence of POMDP-generated actions (Fig. 5), of which the discounted cumulative reward is 57691.7, to complete the *Covered*$_1$ scenario by moving its pose, changing its configuration, declaring, and removing the target object within 14 steps. The robot removes the obstacles (objects 3 and 4) to free the visual occlusion and finally declares and removes the target objects. The sub-images in the red boxes show the probability of the grid world for the guessed target objects, which is updated using Eq. 1, and the ones in yellow boxes show the detected camera images. We can find that, with the growth of the robot's actions, the probability of most grids far away from the target object decreases quickly and the ones near the target object increase a lot, which guides the robot to make more effort to observe the area with the target object.

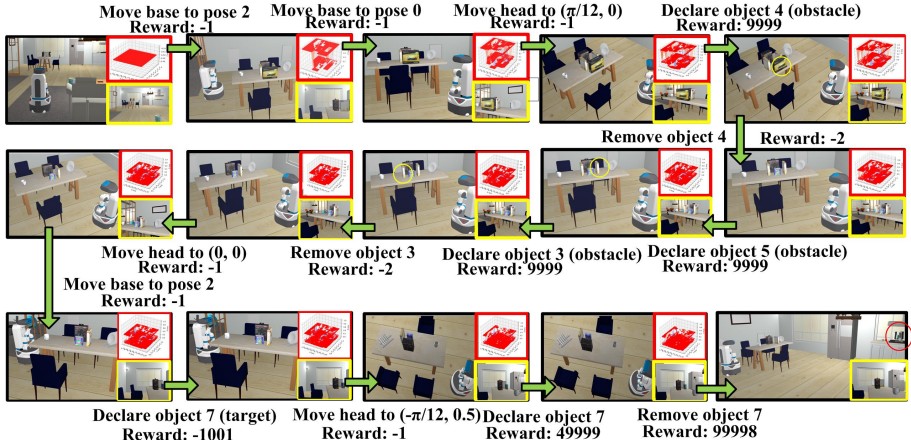

Figure 5: Actions to complete the *Covered*$_1$ scenario with 7 objects in 14 steps.

Our method, utilizing Monte Carlo update, demonstrates superior performance in complex object search scenarios such as cases like *Hidden*$_1$ and *Covered*$_1$, where predicting camera observations is challenging. In simpler cases like *Loose*$_1$, our method performs comparably to the POMCP method when using our introduced guessed target object. The guessed target object proves to be beneficial in activating the robot in situations with limited information. By incorporating our approximated belief tree reuse, we retain more useful branches after receiving observations, leading to further improvement in overall performance. The concept of the guessed target object aligns with our natural tendency to assume the presence of certain objects in specific locations and adjust our estimations based on visual observations.

**Limitation**    Our main limitations mainly stem from the errors and failures from perception and navigation components, not our focused planning part. The first limitation is the

Table 1: Discounted cumulative reward, steps, and successful rate (within 50 steps)

| Scenarios | $Loose_1$ | $Loose_2$ |
|---|---|---|
| POMCP | $66003.7 \pm 16269.5 \mid 13.5 \pm 6.8 \mid 90\%$ | $26175.8 \pm 8220.3 \mid 29.2 \pm 7.7 \mid 65\%$ |
| $POMCP_f$ | $78629.6 \pm 8472.1 \pm \mid 7.2 \pm 0.7 \mid \mathbf{100\%}$ | $38462.6 \pm 11221.2 \mid 23.2 \pm 6.6 \mid 95\%$ |
| $Ours_b$ | $\mathbf{84701.5 \pm 5546.8} \mid \mathbf{6.9 \pm 0.6} \mid \mathbf{100\%}$ | $33265.5 \pm 10085.3 \mid 22.2 \pm 4.2 \mid \mathbf{100\%}$ |
| $Ours_m$ | $79788.0 \pm 4787.4 \mid 7.4 \pm 0.6 \mid \mathbf{100\%}$ | $\mathbf{51574.1 \pm 17930.9} \mid \mathbf{17.4 \pm 2.9} \mid \mathbf{100\%}$ |
| Scenarios | $Hidden_1$ | $Covered_1$ |
| POMCP | $40958.7 \pm 8070.9 \mid 20.1 \pm 2.7 \mid \mathbf{100\%}$ | $17700.2 \pm 3616.6 \mid 38.3 \pm 5.1 \mid 65\%$ |
| $POMCP_f$ | $45815.2 \pm 7260.9 \mid 19.4 \pm 2.8 \mid \mathbf{100\%}$ | $31506.9 \pm 6249.7 \mid 32.3 \pm 6.0 \mid 80\%$ |
| $Ours_b$ | $45038.6 \pm 7631.0 \mid 18.2 \pm 2.7 \mid \mathbf{100\%}$ | $26130.6 \pm 3368.7 \mid 30.2 \pm 4.6 \mid 90\%$ |
| $Ours_m$ | $\mathbf{55574.8 \pm 6225.3} \mid \mathbf{15.3 \pm 2.0} \mid \mathbf{100\%}$ | $\mathbf{34397.8 \pm 7381.6} \mid \mathbf{26.7 \pm 4.8} \mid 95\%$ |

pre-existing map $\mathcal{M}$ which may not be available for a real robot but can be solved by introducing advanced sementic SLAM method. Achieving 100% primitive action success in reality is impractical, affecting our method's success rate. Another limitation is that our framework is based on point cloud segmentation to identify different objects. It may generate incorrect bounding boxes for objects with extensive contact areas. This misinterpretation can result in erroneous data association, impacting belief updates and removing actions. Furthermore, YOLO and SIFT object detection techniques may struggle in scenarios with limited SIFT features and YOLOv5 toolbox coverage, especially in low-light environments. However, we believe that addressing these limitations is possible by incorporating more advanced perception and navigation methods.

## 7   Conclusion and future work

We propose a novel object-level POMDP formulation with growing state space, a guessed target object, and its grid world for object search in 3D with real-world sensors and solve it using a novel GPOMCP method based on Monte-Carlo tree search and belief tree reuse. We demonstrate the effectiveness of our approximation in cases with similar beliefs. Our method achieves object search with only maps, photos, and onboard sensors without relying on additional object information, like geometric models, making it highly practical for real-world robot applications. By utilizing the guessed target object, our method can successfully locate fully hidden objects even without prior information, by removing obstacles—a challenge for existing approaches. Our future directions involve dropping prior information, such as pre-built point cloud map $\mathcal{M}$, and exploring continuous action domains. Through our experiments, we showcase the efficiency of our method, and we anticipate that our results will inspire further advancements in this field.

## Acknowledgements

This research is supported by the ANU Futures Scheme.

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
