# 1 Appendix

## 2 A Additional explanations to the guess target object

3 Our guessed target object is not a real object or a future detected object. The real target object
4 will be included in one of the later sub-state-vector $s_{o^0}$ to $s_{o^n}$. In our problem, initially, we do
5 not know the objects in the environment, neither the number of objects nor their positions,
6 which means the whereabouts of the target object are also initially unknown. To capture
7 uncertainty about the target object compactly, we propose to "guess" the target object. It can
8 be considered as an additional visible object, related to the target object, in transition and
9 visual observation functions of the POMDP framework, but not detectable in the real world.
10 It uses the same structure as the structure used to represent an object in the system, which
11 includes an 8-cell occupancy grid map where each cell maintains a probability of whether
12 the cell intersects with the target object. Under the POMDP framework, a belief over the
13 guessed target object variable represents multiple guesses with different probabilities about
14 the target object. Initially, the uncertainty about the target object can be quite large, but over
15 time the belief in the position of the guess target object tends to converge to a probability
16 mass whose mode is at the true target object. The basic proof of this convergency is shown
17 as follows:

18 *Proof.* Assuming all grids are updated at least $N$ times in the whole action sequence with
19 $N_1 \geq N$ observations and, in each observation $z_t$, any $t \in [0, N_1]$, the values $\log Odd(g_i|z_t)$, $i =$
20 $1, 2, 3, 4$, $g_i \in \mathcal{G}_f$ corresponding to different cases (a. the grid $g_1$ in field of view (FOV)
21 where no objects are detected, b. $g_2$ in the FOV area where non-target objects are detected,
22 c. $g_3$ in FOV where the target object is detected, d. $g_4$ outside the FOV area) are accurate
23 enough to satisfy:

$$\log Odd(g_i|z_t) \begin{cases} < -\triangle < 0 & i = 1, 2 \\ > \triangle > 0 & i = 3 \\ = 0 & i = 4 \end{cases} \tag{1}$$

24 We have:

$$\log Odd(g_i|z_{1:N_1}) \begin{cases} \leq \log Odd(g_i|z_{1:N}) < -N\triangle & i = 1, 2 \\ \geq \log Odd(g_i|z_{1:N}) > N\triangle & i = 3 \\ = 0 & i = 4 \end{cases} \tag{2}$$

25 When $N \to +\infty$, we have:

$$Odd(g_i|z_{1:N_1}) \begin{cases} < \exp^{-N\triangle} \to 0 & i = 1, 2 \\ > \exp^{N\triangle} \to +\infty & i = 3 \\ = 1 & i = 4 \end{cases} \tag{3}$$

26 Introducing the relationship of Odd and probablity values $Odd(g_i|z_{1:N_1}) = \frac{P(g_i|z_{1:N_1})}{1-P(g_i|z_{1:N_1})}$, we
27 have:

$$P(g_i|z_{1:N_1}) = \frac{Odd(g_i|z_{1:N_1})}{1 + Odd(g_i|z_{1:N_1})} \begin{cases} \to 0 & i = 1, 2 \\ \to 1 & i = 3 \\ = 0.5 & i = 4 \end{cases} \tag{4}$$

28 Normalized probabilities (0-1 range) across the grid world lead to convergence towards
29 actual target object grid cells. □

30 The distribution of the guess target object (grid world) saves the belief about the target
31 object. Intuitively, the whole grid world can be used for prediction directly. But, in POMDP
32 problems, we need to predict the future probability of all girds in MCTS by going through
33 the transition functions and observation functions, which will be time-consuming and

memory-consuming based on the whole grid world. The application of the guess target object becomes meaningful to involve the useful information (represent grid world), is very computational (fast for transition functions and observation functions) and memory (a low-dimension vector) cheap in exploration and rollout, and is friendly for coding (use the same transition and visual observation function as the other detected real objects).

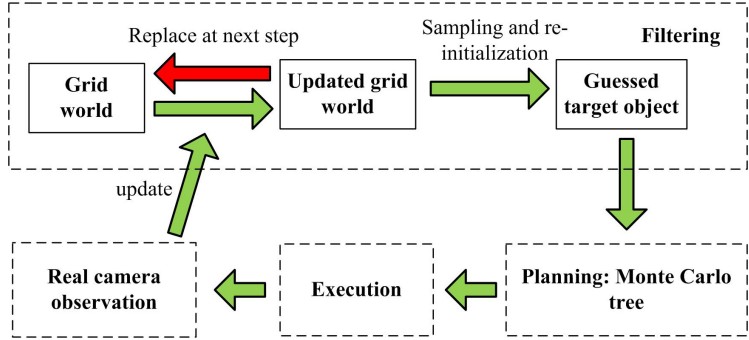

Figure 1: Role of the guess target object

The way to use the guessed target object is shown in Fig. 1. Its grid world is updated based on the real-world observation from the camera and its position will be sampled and the other terms of the guess target object will be re-initialized to use in the transition function and observation function of the Monte Carlo tree search. Then, this iterative process persists until the task is completed.

## B  Additional details to perform the measurement from point cloud before object detection

The main process has been stated in the article. Here, we would like to show a visible process to further explain the main process, including ICP-based scan matching, point cloud filter, point cloud segmentation, and object-oriented bounding box generation, as shown in Fig. 2.

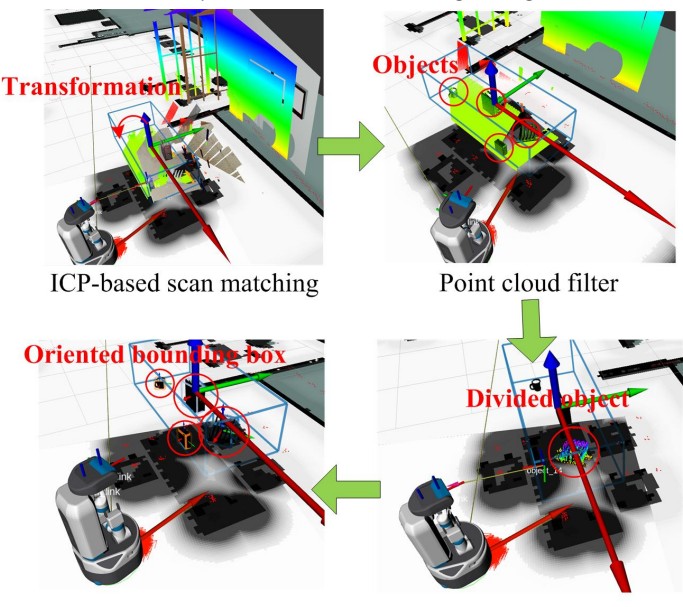

Figure 2: Measurement from point cloud

## C  Object detector

State-of-the-art real-time object detection systems, like YOLO, are commonly designed to divide the object into different classes and they are not matched with the target images. Meanwhile, we have the 3D point clouds of the objects, which are helpful to divide the

objects in the image. So as to complete the given object detection task using several given images and some semantic words (optional), we fuse the traditional feature-matching method and YOLO toolbox to complete the object detection task.

Based on the previous point cloud segmentation, we perform it on the current visual local frame and the separated point clouds in the local frame are re-projected to the image to bound the objects in the RGB camera image forming a set of sub-images $\{\mathcal{I}_i^p, i = 1, 2, \cdots, n\}$ using the camera configuration and the perspective projection. Similar sub-images of this image $\{\mathcal{I}_i^y, i = 1, 2, \cdots, m\}$ and their corresponding semantic scores $\{s_i^y, i = 1, 2, \cdots, m\}$ can also be bounded and generated using the YOLOv5 model with pre-trained parameters. Commonly, we have $m \neq n$. A simple data association method with the nearest images and enough common areas is presented to match these two sets of sub-images. For the successful data association pair, we use the sub-image in the local frame as the image corresponding to this object. These sub-images in the detected and associated 2D boxes corresponding to different objects are matched with the target object using SIFT descriptor. The rate between the number of matched scale-invariant features and the number of all features is defined as the probability of object detection, denoted $\{s_i^d, i = 1, 2, \cdots, n\}$. If this task offers the target type, like cup and laptop, we use the mean values between the semantic scores $\{s_i^y, i = 1, 2, \cdots, n\}$ and the probability of object detection $\{s_i^d, i = 1, 2, \cdots, n\}$. The main process of the object detector is given in Fig. 3.

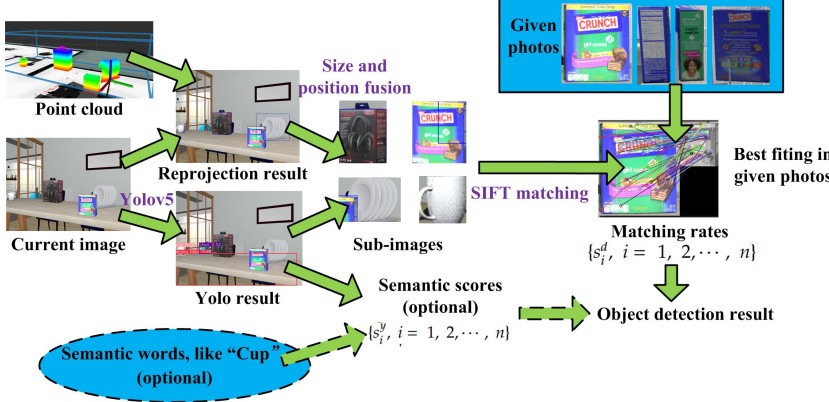

Figure 3: Object detector

## D Move-ability estimation

It is easy to know that, in the real-world environment, some objects in the workspace are not moveable for the robot with a manipulator due to some physical limitations, such as the size limitation of the object, the manipulator workspace limitation, and the mobile base motion limitation. In our framework, we would like to manipulate the objects in the workspace to free some FOV, so it is better to estimate the probability of the move-ability and then update their beliefs for POMDP planning.

Based on the point cloud segmentation for the fused global point cloud, we can obtain many separated point clouds for different objects. Then, facing each point cloud in the detected frame, many candidate grasp poses are predicted by the learning-based Grasp Pose Detection (GPD) toolbox. So as to reduce the computational complexity, we select $k$ representative grasp poses $\boldsymbol{p}_i^g, i = 1, 2, \cdots, k$ for each object using k-means clustering algorithm. These $k$ grasp poses are diverse with high scores in picking success rates. The point clouds of the obstacles in the surrounding environment and these $k$ representative grasp poses $\boldsymbol{p}_i^g$ are transformed to the local frames $T_r^g(\boldsymbol{p}_i^g)$ based on the pre-visited robot poses $\boldsymbol{p}_i^r$ during the task process. Here, it is noted that only the pre-visited robot poses are considered because the poses generated by other methods may not be reachable based on the used move-base toolbox because of the error of the AMCL localization and the complexity of the occupancy grid map. These pre-visited robot poses $\boldsymbol{p}_i^r$ are safer for implementation. Following, these transformed local poses $T_r^g(\boldsymbol{p}_i^g)$ will be set as the plan target to the robot manipulator using moveit toolbox without execution in a given time limitation $t_m$. The

planned moveit feedback will decide the probability of this detection about the move-ability $0 < r_{move} \leq 1$ based on distance. When some objects are near the given poses within a small distance, it will be set as 1 (definitely moveable). When no solution for moveit toolbox, the move-ability $r_{move}$ will get close to 0. In the planning stage, for each particle, we will randomly sample a random value for this object and compare it with the move-ability $r_{move}$ to identify the move-ability in this step. Objects with too large sizes will be considered to be non-moveable $r_{move} = 0$, which is definitely not movable. An example of the candidate grasp poses is shown in Fig. 4.

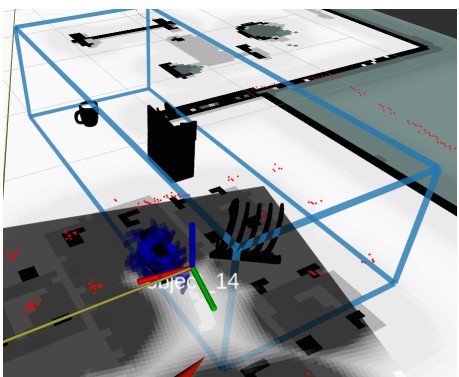

Figure 4: Candidate grasp poses

## E  Theorem 1

**Theorem 1.** *We are considering the unweighted particles for approximating the extended full-correct belief $b(s'_{new} : s' \rightleftarrows s'_{add}) = \frac{1}{N} \sum_{i=1}^{N} \mathbb{I}(s'_{new} = s^i_{new}{}' : s' \rightleftarrows s'_{add})$ and the reused approximated belief $b(s'_{new} : s' \rightarrow s'_{add}) = \frac{1}{N} \sum_{i=1}^{N} \mathbb{I}(s'_{new} = s^i_{new}{}' : s' \rightarrow s'_{add})$. Assuming the reward function is Borel measurable and bounded, denoted $R_{max} = \|R\|_\infty$, the belief L1 distance is limited in $\|b(s'_{new} : s' \rightleftarrows s'_{add}) - b(s'_{new} : s' \rightarrow s'_{add})\|_1 \leq \delta$, and the optimal action selected by building the whole tree is same as the one selected by approximated value $\max_a V(b(s_{new}), a) = \max_a \hat{V}(b(s_{new}), a) = a^*$, the optimal value function of POMDP problem using our method $V^*(b(s_{new}))$ and the optimal value function using the direct resampling way $\hat{V}^*(b(s_{new}))$ will satisfy the following bounding equation:*

$$\|V^*(b(s_{new})) - \hat{V}^*(b(s_{new}))\|_1 \leq \frac{\gamma R_{max}}{1 - \gamma} \delta \tag{5}$$

*Proof.* Let's consider one of the new extended beliefs $b(s'_{new} : s' \rightleftarrows s'_{add})$. Based on the well-known $\alpha$-vector, we have the optimal value of belief $b(s'_{new} : s' \rightleftarrows s'_{add})$ can be written as:

$$
\begin{aligned}
&V^*(b(s'_{new} : s' \rightleftarrows s'_{add})) \\
&= \max_a Q^*(b(s'_{new} : s' \rightleftarrows s'_{add}), a) \\
&= \sum_{s'_{new} \in S} \alpha(s'_{new}) b(s'_{new} : s' \rightleftarrows s'_{add})
\end{aligned} \tag{6}
$$

where $\alpha(s'_{new}) = R(s'_{new}, a^*) + \gamma V^*(s'_{new}, b(s'_{new} : s' \rightleftarrows s'_{add}), a^*)$ and $a^*$ is the optimal action. It is noted that the $\alpha$-vector $\alpha(s'_{new})$ is bounded by $\frac{R_{max}}{1-\gamma}$, introducing the particle representation

115    for the belief $b(s'_{new} : s' \rightleftarrows s'_{add}) = \frac{1}{N} \sum_{i=1}^{N} \mathbb{I}(s'_{new} = s^{i}_{new}{}' : s' \rightleftarrows s'_{add})$, so we have:

$$\|V^*(b(s'_{new} : s' \rightleftarrows s'_{add})) - V^*(b(s'_{new} : s' \rightarrow s'_{add}))\|_1$$

$$= \|\frac{1}{N} \sum_{i=1}^{N} \alpha(s'_{new}) \mathbb{I}(s'_{new} = s^{i}_{new}{}' : s' \rightleftarrows s'_{add}) - \frac{1}{N} \sum_{i=1}^{N} \alpha(s'_{new}) \mathbb{I}(s'_{new} = s^{i}_{new}{}' : s' \rightarrow s'_{add})\|_1$$

$$\leq \|\frac{1}{N} \sum_{i=1}^{N} \frac{R_{max}}{1-\gamma} \mathbb{I}(s'_{new} = s^{i}_{new}{}' : s' \rightleftarrows s'_{add}) - \frac{1}{N} \sum_{i=1}^{N} \frac{R_{max}}{1-\gamma} \mathbb{I}(s'_{new} = s^{i}_{new}{}' : s' \rightarrow s'_{add})\|_1 \qquad (7)$$

$$= \frac{R_{max}}{1-\gamma} \|b(s'_{new} : s' \rightleftarrows s'_{add}) - b(s'_{new} : s' \rightarrow s'_{add})\|_1$$

$$\leq \frac{R_{max}}{1-\gamma} \delta$$

116    Based on

$$\max_{a \in A} \left( \sum_{s_{new} \in S} R(s_{new}, a, s'_{new}) b(s_{new}) + \gamma \sum_{o \in O} \sum_{s'_{new} \in S} Z(s'_{new}, a, o') T(s_{new}, a, s'_{new}) V^*(b(s'_{new} : s' \rightleftarrows s'_{add})) \right)$$

$$(8)$$

117    and the optimal action $a^*$, we have:

$$\|V^*(b(s_{new})) - \hat{V}^*(b(s_{new}))\|_1$$

$$= \left\| \sum_{s_{new} \in S} R(s_{new}, a^*, s'_{new}) b(s_{new}) + \gamma \sum_{o \in O} \sum_{s'_{new} \in S} Z(s'_{new}, a^*, o') \, T(s_{new}, a^*, s'_{new}) V^*(b(s'_{new} : s' \rightleftarrows s'_{add})) \right.$$

$$\left. - \sum_{s_{new} \in S} R(s_{new}, a^*, s'_{new}) b(s_{new}) - \gamma \sum_{o \in O} \sum_{s'_{new} \in S} Z(s'_{new}, a^*, o') \, T(s_{new}, a^*, s'_{new}) V^*(b(s'_{new} : s' \rightarrow s'_{add})) \right\|_1$$

$$= \left\| \gamma \sum_{o \in O} \sum_{s'_{new} \in S} Z(s'_{new}, a^*, o') T(s_{new}, a^*, s'_{new}) \, V^*(b(s'_{new} : s' \rightleftarrows s'_{add})) - \gamma \sum_{o \in O} \sum_{s'_{new} \in S} Z(s'_{new}, a^*, o') \right.$$

$$\left. T(s_{new}, a^*, s'_{new}) V^*(b(s'_{new} : s' \rightarrow s'_{add})) \right\|_1$$

$$\leq \gamma \sum_{o \in O} \sum_{s'_{new} \in S} Z(s'_{new}, a^*, o') T(s_{new}, a^*, s'_{new}) \frac{R_{max}}{1-\gamma} \delta$$

$$= \frac{\gamma R_{max}}{1-\gamma} \delta$$

$$(9)$$

118    The proof is completed.                      $\square$

## 119   F    Corollary 2

120   **Corollary 1.** *If the distance between the value function for the optimal action $V(b(s_{new}), a^*)$ and the*
121   *value function for any sub-optimal action $V(b(s_{new}), a^*_{sub})$ is larger than $\frac{2\gamma R_{max}}{1-\gamma}\delta$, the optimal action*
122   *obtained by the tree reuse approximation way will be same as the optimal action using the direct*
123   *resampling way $\max_a V(b(s_{new}), a) = \max_a \hat{V}(b(s_{new}), a) = a^*$.*

124   *Proof.* Based on Corollary 1, we have:

$$\hat{V}(b(s_{new}), a^*_{sub}) \leq V(b(s_{new}), a^*_{sub}) + \frac{\gamma R_{max}}{1-\gamma} \delta \qquad (10)$$

125   and

$$V(b(s_{new}), a^*) - \frac{\gamma R_{max}}{1 - \gamma}\delta \leq \hat{V}(b(s_{new}), a^*) \tag{11}$$

The solution does not change, which means that the approximated value function using sub-optimal action $a_{sub}$ is smaller than the one with optimal action, satisfying $\hat{V}(b(s_{new}), a^*) > \hat{V}(b(s_{new}), a_{sub}^*)$. Here, considering Eq. (13) and Eq. (12), if the upper bound of $\hat{V}(b(s_{new}), a_{sub}^*)$ is smaller than the lower bound of $\hat{V}(b(s_{new}), a^*)$, satisfying:

$$V(b(s_{new}), a_{sub}^*) + \frac{\gamma R_{max}}{1 - \gamma}\delta < V(b(s_{new}), a^*) - \frac{\gamma R_{max}}{1 - \gamma}\delta$$
$$\Rightarrow V(b(s_{new}), a^*) - V(b(s_{new}), a_{sub}^*) > \frac{2\gamma R_{max}}{1 - \gamma}\delta, \tag{12}$$

We have:

$$\hat{V}(b(s_{new}), a_{sub}^*) < \hat{V}(b(s_{new}), a^*) \tag{13}$$

The proof is completed. □

## G   Solver summary

The whole online GPOMDP solver for our object search is shown in the following pseudocode Algorithms 1 to 7. Our proposed method follows the common procedure with four alternating stages, planning, execution, obtaining observation, and filtering shown in Algorithms 1. The optimal action for the current belief is selected based on my solver proposed in Section 5 with belief tree reuse shown in Algorithms 2. Two key points should be pointed out. The first one is that, because of the observation introduced by added objects, after the state update in Section 5 Step 2, some observation nodes will be decoupled into multiple new observation nodes. Hence, the observation identification needs to be performed in line 14 Algorithm 3. The other point is that we save the observation in two hash tables, one from observation to identification ID and the other one from identification ID to observation.

---

**Algorithm 1** Proposed GPOMDP method with growing state space

---

**Input:** POMDP 8-tuple $< \mathcal{S}, \mathcal{A}, \mathcal{O}, T, Z, R, b_0, \gamma >$: state space $\mathcal{S}$, action space $\mathcal{A}$, observation space $\mathcal{O}$, transition function $T(s, a, s')$, observation function $O(s', a, o)$, reward function $R(s, a, s')$, initial belief $b_0$, and discount factor $\gamma$; and Communicable Robot.
**Output:** Optimal action sequence for this POMDP problem.
  1: Sampling initial state $\{s_i\}_{i=1, 2, \cdots, N}$ based on initial belief $b_0$, where is generated based on grid world of the fake object.
  2: **while** True **do**
  3:     $a^* \leftarrow PLANNING(b)$
  4:     Communicate with the robot, execute $a^*$, and reach an unknown state $s_{unknown}$
  5:     $o \leftarrow OBSERVATION(s_{unknown})$
  6:     Update the belief $b$ with $a^*$ and $o$ using Filters
  7:     **if** The object searching task is completed **then**
  8:         **break**
  9:     **end if**
 10: **end while**

---

**Algorithm 2** *PLANNING($b$)*

**Input:** Current belief $b$
**Output:** Optimal action $a^*$ in this step
1: **if** New objects $s'_{add}$ are detected and old tree $\mathcal{T}$ exists **then**
2:    Cut some history from history List $H$ based on previous optimal action $a^*$ and obtained observation $o$
3:    Extend the action space $a_{add}$ for the newly detected objects including removing and declaring actions
4:    **for all** particles in parallel **do**
5:        $s'_{add} \leftarrow T(s_{add}, a)$
6:        $o_{add} \leftarrow Z(s'_{add}, a)$
7:        $H' \leftarrow H \cup \{(s', s'_{add}), a, o_{add}\}$
8:    **end for**      ▷ Update the state vector and observations of the particles considering the newly detected objects. Here we need to consider the effect from the old object to the newly detected object but do not consider the reverse effect. $b(s'_{new} : s' \rightarrow s'_{add})$
9:    **for** all particles within updated history list $\{h'\} \in H'$ **do** ▷ Generate the belief tree $\mathcal{T}'(s)$ using the updated history List $H'$.
10:        $SIMULATE\_NEW\_OBJECTS(H', \varnothing, particle, depth)$
11:    **end for**
12: **end if**
13: **while** time permitting or particle number limitation **do** ▷ Sampling more particles for MCTS if the reuse tree operation does not spend too much planning time.
14:    Sampling a state $s_{new}$ based on belief $b$
15:    $SIMULATE(s_{new}, \varnothing, depth)$
16: **end while**
17: **return** $\arg\max_{a \in \mathcal{A}} \hat{Q}(b, a)$

---

**Algorithm 3** *SIMULATE_NEW_OBJECTS($H'$, $h$, $i$, $j$)*

**Input:** The record of all history $H'$ with saved states, actions, observations, and reward; The particle id $i$; The depth $j$
**Output:** Discounted total reward $r$
1: $h' \leftarrow H'(i, 0 : j)$                                                           ▷ Follow the old history.
2: **if** $j == |h'|_{rollout}$ **then**
3:    **if** Not Action node $T(h' + \{a\})$ **then**
4:        **for all** $a \in \mathcal{A}$ **do**
5:            Action node $T(h' + \{a\}) \leftarrow (N(h' + \{a\}), V(h' + \{a\}), \varnothing)$
6:        **end for**
7:    **end if**
8:    **return** $ROLLOUT\_NEW\_OBJECTS(h', j + 1)$
9: **else**
10:    **if** not $T(h' + \{a\})$ **then**
11:        **for all** $a \in \mathcal{A}$ **do**
12:            Action node $Node(h' + \{a\}) \leftarrow (N(h' + \{a\}), V(h' + \{a\}), \varnothing)$
13:        **end for**
14:        Action node $Node(h' + \{a\}) \leftarrow (N(h' + \{a\}), V(h' + \{a\}), \varnothing)$
15:    **end if**
16:    $\{s_{new}, s'_{new}, a, o_{add}, R(s_{new}, a, s'_{new})\} \leftarrow H'(i, j)$
17:    **if** $j < |h'|_{rollout}$ **then**
18:        $r \leftarrow \gamma SIMULATE\_NEW\_OBJECTS(H', h + \{a, o_{add}\}, i, j + 1) + R(s_{new}, a, s'_{new})$
19:        $B(h') \leftarrow B(h') \cup \{s_{new}\}$
20:        $N(h') \leftarrow N(h') + 1$
21:        $N(h' + \{a\}) \leftarrow N(h' + \{a\}) + 1$
22:        $\hat{Q}(h' + \{a\}) \leftarrow \hat{Q}(h' + \{a\}) + \frac{r - \hat{Q}(h' + \{a\})}{N(h' + \{a\})}$              ▷ Monte Carlo update.
23:    **end if**
24: **end if**

---

## H  Discussion about different resolution for grid world

The grid world's finer resolution offers a unique advantage in enhancing the reliability of probability updating within the grid. This advantage becomes particularly pronounced when dealing with smaller target objects. It's important to acknowledge that opting for this finer resolution does entail a slightly higher computational complexity (odds updating is

**Algorithm 4** *SIMULATE*($\boldsymbol{s}$, $h$, *depth*)

**Input:** A given state $\boldsymbol{s}_{new}$, the previous history $h$, the node depth *depth*
**Output:** Discounted total reward $r$
1: Observation node $Node(h) \leftarrow (N(h), V(h), depth)$
2: **if** Not $Node(ha)$ **then**
3:     **for all** $\boldsymbol{a} \in \mathcal{A}$ **do**
4:         Action node $Node(h + \{\boldsymbol{a}\}) \leftarrow (N(h + \{\boldsymbol{a}\}), V(h + \{\boldsymbol{a}\}), depth)$
5:     **end for**
6:     **return** $ROLLOUT(h', j + 1)$
7: **else**
8:     $\boldsymbol{a} \leftarrow \text{argmax}_{\boldsymbol{a}} C_b \sum_{o \in O} Z(\boldsymbol{s}'_{new}, \boldsymbol{a}, \boldsymbol{o}') \|\boldsymbol{b}((\boldsymbol{s}', \boldsymbol{s}'_{add}) : s' \rightarrow \boldsymbol{s}'_{add}) - \boldsymbol{b}((\boldsymbol{s}', \boldsymbol{s}'_{add}) : s' \rightleftarrows \boldsymbol{s}'_{add})\|_1 + V^*(\boldsymbol{b}, \boldsymbol{a}) +$
    $C_N \sqrt{\frac{logN(b)}{N(\boldsymbol{b}, \boldsymbol{a})}}$                       ▷ Novel UCB strategy.
9:     $\boldsymbol{s}'_{new} \leftarrow T(\boldsymbol{s}_{new}, \boldsymbol{a})$
10:    $\boldsymbol{o}' \leftarrow Z(\boldsymbol{s}'_{new}, \boldsymbol{a})$
11:    **if** $\boldsymbol{s}'_{new}$ is not terminal state **then**
12:       $r \leftarrow \gamma SIMULATE(\boldsymbol{s}_{new}, h + \{\boldsymbol{a}, \boldsymbol{o}_{add}\}, depth + 1) + R(\boldsymbol{s}_{new}, \boldsymbol{a}, \boldsymbol{s}'_{new})$
13:       $B(h') \leftarrow B(h') \cup \{\boldsymbol{s}_{new}\}$
14:       $N(h') \leftarrow N(h') + 1$
15:       $N(h' + \{\boldsymbol{a}\}) \leftarrow N(h' + \{\boldsymbol{a}\}) + 1$
16:       $\hat{Q}(h' + \{\boldsymbol{a}\}) \leftarrow \hat{Q}(h' + \{\boldsymbol{a}\}) + \frac{r - \hat{Q}(h' + \{\boldsymbol{a}\})}{N(h' + \{\boldsymbol{a}\})}$         ▷ Monte Carlo update.
17:    **end if**
18: **end if**

---

**Algorithm 5** *ROLLOUT_NEW_OBJECTS*($h'$, $j$)

**Input:** The history $h$, the depth $j$
**Output:** Discounted total roll-out reward
1: **if** $j > |h'|$ **then**
2:     **return** $0$
3: **end if**
4: $\{\boldsymbol{s}_{new}, \boldsymbol{s}'_{new}, \boldsymbol{a}, \boldsymbol{o}_{add}, R(\boldsymbol{s}_{new}, \boldsymbol{a}, \boldsymbol{s}'_{new})\} \leftarrow h'(j)$
5: **if** $j == |h'|$ **then**
6:     **return** $R(\boldsymbol{s}_{new}, \boldsymbol{a}, \boldsymbol{s}'_{new})$
7: **end if**
8: **return** $r + \gamma ROLLOUT\_NEW\_OBJECTS(h' + \{\boldsymbol{a}, \boldsymbol{o}_{add}\}, j + 1)$

---

**Algorithm 6** *ROLLOUT*($\boldsymbol{s}, h, j$)

**Input:** The history $h$, the depth $j$, the current state $s$
**Output:** Discounted total roll-out reward
1: **if** $\gamma^j < C_r$ or $j > C_{max}$ **then** ▷ Update the state vector and observations of the particles considering the newly detected objects. Here we need to
2:     **return** $0$
3: **end if**
4: $\boldsymbol{a} \leftarrow random(\mathcal{A})$
5: $\boldsymbol{s}' \leftarrow T(\boldsymbol{s}, \boldsymbol{a})$
6: $\boldsymbol{o} \leftarrow Z(\boldsymbol{s}', \boldsymbol{a})$
7: $r \leftarrow R(\boldsymbol{s}, \boldsymbol{a}, \boldsymbol{s}')$
8: **return** $r + \gamma ROLLOUT(\boldsymbol{s}, h + \{\boldsymbol{a}, \boldsymbol{o}\}, j + 1)$

---

155  very cheap for large datasets). Upon conducting tests, it becomes evident that the good
156  grid resolution should be smaller than the minimum dimensions of the target object along
157  both x and y axes. We are here to correct a typo in our previous version. All the experi-
158  ments are implemented in 2 cm resolution instead of 5 cm. For comparison, we present
159  statistical results for the scenario using 10 cm resolution: 24803.5±3792.3|39.9±6.9|60%.
160  The result is poorer than the POMCP method without using fake objects in success
161  rate. To dissect this phenomenon, we manually executed a designated sequence of ac-
162  tions: ⸢$move\_head\_5 - move\_base_0 - move\_base\_1 - move\_base\_2 - move\_base\_3 - move\_lift\_2 -$
163  $move\_base\_0 - move\_base\_1 - move\_base\_2 - move\_base\_3 - remove\_object\_4 - move\_head\_8 -$
164  $move\_base\_0 - move\_base\_1 - move\_base\_2 - move\_base\_3$⸥ for our method with 2 cm and 10 cm
165  resolution, and the resulting changes in probability distributions across the grid world are
166  visually depicted in Fig. 5. The salient observation is that when the resolution is relatively

**Algorithm 7** $o \leftarrow OBSERVATION(s_{unknown})$

---

**Input:** Current configuration including robot and environment
**Output:** Obtained observation for pose and objects
1: ICP-based scan matching to get the rigid transformation $T_r$ between AMCL pose estimation and 3D point cloud map
2: Point cloud fusion $\mathcal{P}_j = \mathcal{M} \bigcup \mathcal{F}_0 \bigcup \cdots \bigcup \mathcal{F}_j$
3: Point cloud filter to remove the point cloud outside the workspace
4: Point cloud segmentation to divide the point cloud into multiple point clouds $\{o_0, o_1, \cdots, o_n\} \in \mathcal{M}'_j$
5: Minimal oriented bounding box estimation for the poses $\{p_0^o, p_1^o, \cdots, p_n^o\} \in \mathcal{M}'_j$ of different point clouds
6: SIFT and YOLO-based object detector with detected scores $\{(s_i^y + s_i^d)/2, i = 1, 2, \cdots, n\}$
7: Move ability estimation based on GPD toolbox
8: Data association and summarize all measurements, including robot pose estimation, object pose and size estimation, probability of object detection (most important, only used in the belief tree), and move-ability estimation

---

167 large, grid probability updates lose precision, particularly when the target object is situated
168 close to an obstacle. The use of a larger resolution not only fails to enhance our method's
169 performance but also introduces potential distortions in the information of the fake object.
170 In short, we recommend the used resolution had better be smaller than the object sizes to
171 improve the accuracy of updating odds values.

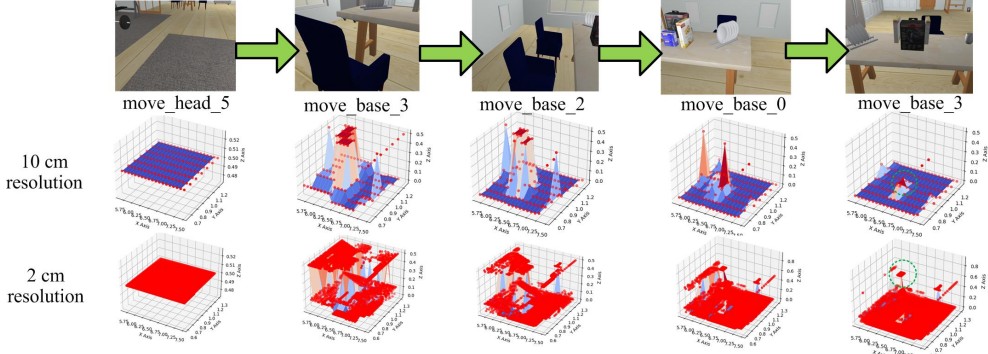

Figure 5: Candidate grasp poses

## I More simulation results with different object numbers, reward settings, and thresholds

174 Indeed, the computational complexity of our framework will mainly increase with the
175 number of updating objects. These objects play a crucial role in both transition and visual
176 observation functions, causing fewer sampled particles. Consequently, the computational
177 expenditure associated with these declared objects becomes relatively economical, as it no
178 longer necessitates exhaustive testing of occlusion relationships among all other objects. The
179 good news is that commonly the number of the updating objects is limited by the declaring
180 action. To facilitate a clearer understanding of how our method performs under varying
181 object counts, we have conducted an ablation study. This study includes a comprehensive
182 set of experiments, encompassing an increasing number of objects ranging from 2 to 10, with
183 increments of 2. The scenarios, as depicted in Fig. 6, comprise 20 trials each, all adhering to
184 a consistent planning time limit of 60 seconds per step. The statistical results are reported in
185 Fig. 7. Our method will perform obviously better, if the object number is relatively large
186 (blue box areas), because we reuse the useful belief tree and avoid branch-cutting caused by
187 the newly detected objects, which is more common in scenarios with more objects.

188 While some parameters of our framework require manual configuration, it's worth noting
189 that the majority of these parameters do not significantly impact the final performance of
190 the framework. Most of them are not sensitive to the final performance. As an example, the
191 performance does not change too much, if orders of magnitude between $R_{max}$ and $R_{min}$ satisfy
192 $R_{max} \gg R_{min}$. For instance, $R_{min}$ varying from -1 to -20 does not exert a substantial influence

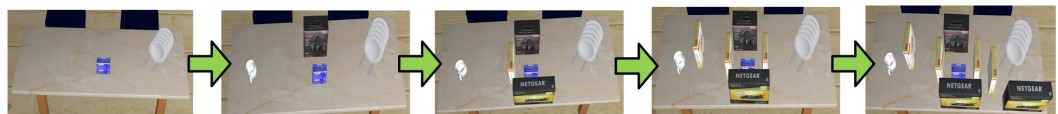

**Object number = 2    Object number = 4    Object number = 6    Object number = 8    Object number = 10**

Figure 6: Scenarios with different object numbers from 2 to 10.

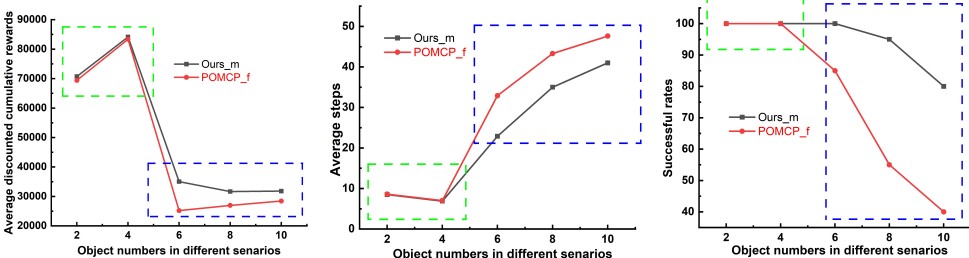

(a) Average discounted cumulative rewards (larger is better)  (b) Average steps (fewer is better)  (c) Successful rate (higher is better)

Figure 7: The comparisons between Ours_m with POMCP_f.

on performance outcomes. To elucidate this phenomenon, we present a comparison between the results using $R_{min} = -20$ (the case in our paper) and for the $Hidden_1$ case in Table 1.

Table 1: 95% confidence interval of discounted cumulative reward, steps, and successful rate (within 50 steps)

| Scenarios | $R_{min} = -1$ | $R_{min} = -20$ |
|---|---|---|
| $Hidden_1$ | $55574.8 \pm 6225.3 \mid 15.3 \pm 2.0 \mid 100\%$ | $54351.1 \pm 7180.3 \mid 15.9 \pm 2.2 \mid 100\%$ |

Few parameters may affect the performance a lot. For example, before successfully declaring, we need to update the belief of the log-odds of 8 grid values and then complete the declaring action by comparing the mean value of the minimal $n_{odds} = 2$ log-odds with the thresholds $C_d^o$ for the obstacle object and $C_t^o$ for the target object. The smaller $n_{odds}$ means that this condition is hard to be satisfied and we need to observe more directions of the objects. As $n_{odds}$ decreases, the removing and declaration actions become more dependable, owing to the augmented array of diverse observations derived from different orientations. Our method retains more useful branches after receiving observations and it will show obvious advantages when the target belief is harder to reach. With this goal, we increase $n_{odds}$ to 4 and 6 and show the statistical results of the scenario with 6 objects (Fig. 6) in Table 2 with 20 trials.

Table 2: 95% confidence interval of discounted cumulative reward, steps, and successful rate (within 50 steps)

| Scenarios | $Ours_b$ | $POMCP_f$ |
|---|---|---|
| 2 | $\mathbf{35038.7 \pm 4604.5} \mid \mathbf{22.9 \pm 2.5} \mid \mathbf{100\%}$ | $25192.4 \pm 3045.1 \mid 32.9 \pm 3.5 \mid 85\%$ |
| 4 | $45327.6 \pm 4491.9 \mid \mathbf{19.3 \pm 1.8} \mid \mathbf{100\%}$ | $\mathbf{47258.1 \pm 6185.9} \mid 20.2 \pm 3.8 \mid 90\%$ |
| 6 | $\mathbf{72967.1 \pm 10896.3} \mid 12.1 \pm 2.8 \mid \mathbf{100\%}$ | $72081.7 \pm 9524.8 \mid \mathbf{12.0 \pm 2.7} \mid \mathbf{100\%}$ |