# OpenReview forum: "POMDP Planning for Object Search in Partially Unknown Environment"
_NeurIPS.cc/2023/Conference — NeurIPS 2023 poster_

### Official Review · Reviewer_2WUu · 2023-07-03

**Soundness:** 3 good
**Presentation:** 3 good
**Contribution:** 3 good
**Rating:** 7
**Confidence:** 3

**Summary:**

This paper addresses the problem of finding efficient policies for a robot to search for a known target object in a physical environment.  The robot can find the target object by either moving around to change its field of view or by moving objects that occlude the target object to reveal the target object. While the robot has access to 3D point clouds and occupancy grid maps of the environment and included furniture, objects on the furniture are not included in the occupancy grid maps.   The authors frame this as a POMDP as the state of the environment is partially observed.  The authors propose a novel POMDP solver – Growing Partially Observable Monte-Carol Planner (GPOMCP) – which reuses the previous belief tree under very specific conditions as defined in Theorem 1, Corollary 1,2 – where the optimal action obtained by reusing the tree will be the same as direct sampling.  Overall this is an important practical problem for robots and a new technique for addressing the infeasibility of exactly solving the resulting POMDP is welcome.

I appreciate the responses of the authors during the rebuttal phase and have raised my rating to Accept.

**Strengths:**

The authors clearly explain their approach for action execution using ROS interfaces move-base, ros_control and moveit, perception for object detection using point cloud segmentation, point cloud re-projection, sub-image fusion, YOLO detection and SIFT matching, and an estimation of each objects move-ability using the Grasp Pose Detection toolbox, k-means clustering and moveit. The authors explain their planning method, GPOMCP, including their use of the belief tree, and provide appropriate steps, equations, theorems, and corollaries. The authors present simulation results using Gazebo across four environment scenarios based on whether the target object is loose, hidden, or covered and compare their proposed method with the Bellman update (method 1) and the Monte Carlo update (method 2) with the primary POMCP method without the fake object (method 3) and with the fake object (method 4).

The authors also introduce the concept of a “fake object” (last paragraph of section 1), which represents a fully occluded object to facilitate creating a distribution for the pose of this object.  The authors present results from a Gazebo simulation system comparing their algorithm against prior baselines in 4 levels of difficulty based on the visibility of the target object.  Table 1 indicates that the GPOMCP Planner more successfully locates the target object and with fewer steps.

**Weaknesses:**

This paper addresses an interesting problem but the Prior Work section misses important precedents.  In line 105, the authors state that when the target object is fully occluded, this presents a new challenge to the best of their knowledge.  However, this problem – where the target object is fully occluded – is well-known in the robotics literature where it is often referred to as “Mechanical Search”.   The problem has also been posed as a POMDP and addressed with Monte-Carlo Search Tree methods (see references below, 1 uses MC Trees).

The authors did not include a Limitations section, this must be fixed if the paper is to be accepted.

The paper would be improved with more carefully proofreading to make the English clearer, for example the term “fake object” seems inappropriate.  A better term might be “hidden object” as there is nothing “fake” about this, and a more detailed description of how the distribution evolves in the experiments under the “Covered” condition.

Potentially Relevant Prior Work:

Huang, H. et al. (2023). Mechanical Search on Shelves with Efficient Stacking and Destacking of Objects. In: Billard, A., Asfour, T., Khatib, O. (eds) Robotics Research. ISRR 2022. Springer Proceedings in Advanced Robotics, vol 27. Springer, Cham. https://doi.org/10.1007/978-3-031-25555-7_14
Danielczuk, M., et al.: Mechanical search: multi-step retrieval of a target object occluded by clutter. In: 2019 International Conference on Robotics and Automation (ICRA), pp. 1614–1621. IEEE (2019)
Kurenkov, A., et al.: Visuomotor mechanical search: learning to retrieve target objects in clutter. In: 2020 IEEE/RSJ International Conference on Intelligent Robots and Systems (IROS), pp. 8408–8414. IEEE (2020)
Yang, Y., Liang, H., Choi, C.: A deep learning approach to grasping the invisible. IEEE Robot. Autom. Lett. 5(2), 2232–2239 (2020)
Bejjani, W., Agboh, W.C., Dogar, M.R., Leonetti, M.: Occlusion-aware search for object retrieval in clutter. In: 2021 IEEE/RSJ International Conference on Intelligent Robots and Systems (IROS), pp. 4678–4685. IEEE (2021)
Gupta, M., R¨uhr, T., Beetz, M., Sukhatme, G.S.: Interactive environment exploration in clutter. In: 2013 IEEE/RSJ International Conference on Intelligent Robots and Systems, pp. 5265–5272. IEEE (2013)
Huang, H., et al.: Mechanical search on shelves using lateral access x-ray. In: 2021 IEEE/RSJ International Conference on Intelligent Robots and Systems (IROS), pp. 2045–2052. IEEE (2021)
Chen, L.Y., Huang, H., Danielczuk, M., Ichnowski, J., Goldberg, K.: Optimal shelf arrangement to minimize robot retrieval time. In: 2022 IEEE 18th International Conference on Automation Science and Engineering (CASE), pp. 993–1000. IEEE (2022)
Nakhimovich, D., Miao, Y. and Bekris, K.E., 2023. Resolution Complete In-Place Object Retrieval given Known Object Models. arXiv preprint arXiv:2303.14562.

**Questions:**

In lines 53-54, the authors claim that the method will always converge to a delta distribution at the correct target pose.  It would be helpful to justify with a proof, or consider conditions where the method may not converge or may converge to the incorrect object pose.

The paper would also be improved with more details on the specific experiments as Figure 4 is difficult to follow (is the blue box the target in all 4 examples? This is never stated).

On a minor note, the paper has some typos or style choices that need correction, such as:
Line 68: ‘an action’ not ‘a action.’
Line 145: ‘shows’ not ‘to show .’
Line 147: ‘and’ not ‘and but .’
Line 290: ‘rebuilt’ not ‘rebuild’

The authors did not include a Limitations section, this must be fixed if the paper is to be accepted.

**Limitations:**

The authors did not include a Limitations section.

---

> ### Author Rebuttal · Authors · 2023-08-08
>
> $\textbf{Response about weakness}$:
>
> W1. This paper...
>
> $\textbf{Answer}$: We appreciate the reviewer's insightful feedback regarding the Prior Work section. Apologies for overlooking the concept of 'Mechanical Search' in our initial version. We have revised the paper and connected our framework with “Mechanical Search”. Some related papers have been added in the revised version, especially for the concept-related paper (Mechanical search: multi-step retrieval of a target object occluded by clutter) and the MCTS-based paper (Mechanical Search on Shelves with Efficient Stacking and Destacking of Objects).
>
> W2. The authors... and Q4. The authors...
>
> $\textbf{Answer}$: Thanks. Please refer to the “Limitation” section of the global comments file for more details.
>
> W3. The paper...
>
> $\textbf{Answer}$: Thanks. Please refer to the “Fake object” section.
>
> The distribution evolution in the $Covered_1$ case follows a similar pattern to other scenarios. Initially, a uniform distribution spreads across the grid (The odds is 1 for each grid), followed by odds updates using Eq. 1 inspired by occupancy grid map updates adapting with field of view (FOV) changes:
>
> • In FOV where no objects are detected, the log-odds value is incremented by a negative value, indicating that the grid is unlikely to contain the target object.
>
> • In FOV where non-target objects are detected, the log-odds value is augmented by a negative value corresponding to the object detection confidence (<0.5), signifying that the grid is less likely to contain the target object.
>
> • In target object-detected FOV areas, log-odds rise positively with confidence (>0.5), signifying higher target likelihood.
>
> To provide a clearer understanding of the odds updating process in the $Covered_1$ case, we manually executed a designated sequence of actions: “move_head_5 -move_base_0-move_base_1-move_base_2-move_base_3-move_lift_2-move_base_0-move_base_1-move_base_2-move_base_3-remove_object_4-move_head_8-move_base_0-move_base_1-move_base_2-move_base_3” for our method with 2 cm (We are here to correct a typo in our previous submission. All the experiments are implemented in 2 cm resolution instead of 5 cm.) and 10 cm resolution, and Fig. 3 shows resulting grid world probability distribution changes.
>
> In the last two images, locate two green dotted circles. For the one with a 10 cm resolution, the updated probability corresponding to the target object is not predicted well, which is a near-0 value, after all actions. Our 2 cm method offers better prediction (~1), benefiting from finer object resolution. This detail aids in updating reliability, especially for small targets. This advantage becomes particularly pronounced when dealing with smaller target objects, but with slightly higher computation (but odds updating is very cheap commonly). The salient observation is that when the resolution is relatively large, grid probability updates lose precision, particularly when the target object is situated close to an obstacle. The use of a larger resolution not only fails to enhance our method's performance but also introduces potential distortions in the information of the fake object. As evidence, we present statistical results for the covered scenario using a 10 cm resolution: 24803.5±3792.3|39.9±6.9|60%. It is worse than the POMCP without using fake object in success rate. Recommendation: Resolution < object sizes for accurate odds updating.
>
> $\textbf{Response to questions}$:
>
> Q1. In lines 53-54...
>
> $\textbf{Answer}$: Thanks. Apologies for the seemingly overly strong claim, especially in the context of the delta distribution. For instance, when dealing with multiple target objects situated at different positions, the resulting distribution may not exhibit the characteristics of a delta distribution. I've adjusted the claim: “As new accurate observations are perceived, the robot refines its belief about the fake objects, and eventually the belief has a high chance to converge to the correct target object positions based on suitable parameter settings.” However, it is our intention to demonstrate that, in general, the probability tends to behave in a certain manner.
>
> Assuming all grids are updated at least $N$ times in the whole action sequence with $N_1>=N$ observations and, in each observation $z_t$, any $t\in[0, N_1]$, the values $\log Odd(g_i|z_t), i=1,2,3,4$ corresponding to different cases (a. the grid $g_1$ in FOV where no objects are detected, b. $g_2$ in the FOV area where non-target objects are detected, c. $g_3$ in FOV where the target object is detected, d. $g_4$ outside the FOV area) are accurate enough satisfying $\log Odd(g_i|z_t)<-∆<0, i = 1,2$, $\log Odd(g_i|z_t)>∆>0, i=3$, and $\log Odd(g_i|z_t)=0, i=4$. We have:
> $\log Odd(g_i|z_{1: N_1})<= \log Odd(g_i|z_{1: N})<-N∆, i=1,2$ and $\log Odd(g_i|z_{1:N_1}) >=\log Odd(g_i|z_{1:N})>N∆, i=3$, so we have:
> $Odd(g_i|z_{1:N_1})<\exp^{-N∆}, i=1,2$, and $Odd(g_i|z_{1:N_1})>\exp^{N∆}, i=3$,
> If $N→+∞$, $Odd(g_i|z_{1:N_1})→0$ and $P(g_i|z_{1: N_1})= Odd(g_i|z_{1:N_1})/(1+Odd(g_i|z_{1:N_1}))→0, i=1,2$ and  $Odd(g_i | z_{1:N_1})→+∞$ and $P(g_i|z_{1:N_1})→1, i=3$. Normalized probabilities (0-1 range) across the grid world lead to convergence towards actual target object grid cells. This progressive refinement of probabilities contributes to a more precise understanding.
>
> Q2. The paper...
>
> $\textbf{Answer}$: Thanks. We've added sentences clarifying the target object and other specifics, including “The comparison is implemented based on 4 different scenarios with different object numbers, including LOOSE_1 (4 objects in 1 workspace), LOOSE_2 (6 objects in 2 workspaces), Hidden_1 (7 objects in 1 workspace), and Covered_1 (7 objects in 1 workspace),  as shown in Fig. 4. The target object is the blue snack box. Only the black headset box and the dish rack are not movable, and others are all movable.”
>
> Q3. On a...
>
> $\textbf{Answer}$: Thanks for your thorough review. Typos have been corrected in the revision.

---

> > ### Comment · Reviewer_2WUu · 2023-08-16
> > **Thank you for Authors Response to my Review**
> >
> > I appreciate the authors' response to my review.  I have also carefully read the other reviews and responses.  I appreciate the clarifications and thoughtful Limitations sections that the authors will add to the final paper.  I am willing to increase my rating from Borderline Accept to Accept but am very open to the discussion with all reviewers in the next phase.

---

> > > ### Author Response · Authors · 2023-08-16
> > > **Appreciation for Your Responses and Possible Score Update**
> > >
> > > Thanks for your kind feedback on possible score updating. We will continue to improve our final version and try to include all the discussed content in our last version, especially for the Limitation Section (main document) and ablation study (supplementary material). Please feel free to let us know if you have any questions or further suggestions.

---

### Official Review · Reviewer_pj6p · 2023-07-06

**Soundness:** 2 fair
**Presentation:** 2 fair
**Contribution:** 2 fair
**Rating:** 4
**Confidence:** 1

**Summary:**

This paper presents a POMDP based approach for finding objects in partially unknown environments.  They present a planning algorithm that reuses the belief tree and uses a fake object to guide the search.  The system is tested in four simulated scenes in Gazebo and outperforms a POMCP baseline.

**Strengths:**

 - The proposed system achieves better performance than the baseline in several experiments.

**Weaknesses:**

- The main experiment of the paper is a set of comparisons to POMCP [13], which was published in 2010. The related works section of the proposed work mentions several other more modern approaches.  It is not clear why these works were not also compared to.

**Questions:**

- Are the object positions the same in the 20 trials for each scenario or are they randomized?  If they are the same, only testing on four different configurations of objects is too few, especially when the experiments are done in simulation.


- I am confused about how the fake object is used and what the intuition behind it is.

**Limitations:**

Limitations are only very briefly discussed in the conclusion.

---

> ### Author Rebuttal · Authors · 2023-08-08
>
> $\textbf{Response about weakness}$:
>
> W1. The main experiment of the paper is a set of comparisons to POMCP [13], which was published in 2010. The related works section of the proposed work mentions several other more modern approaches. It is not clear why these works were not also compared to.
>
> $\textbf{Answer}$: Thank you very much for this comment. Please refer to the “Baseline methods” section of the global comments file for more details.
>
> $\textbf{Response to questions}$:
>
> Q1. Are the object positions the same in the 20 trials for each scenario or are they randomized? If they are the same, only testing on four different configurations of objects is too few, especially when the experiments are done in simulation.
>
> $\textbf{Answer}$: Thanks for your comments. Yes, they are the same, not randomized. However, we like to point out that in the area of this paper’s main contribution (online POMDP solving and its application in robotics systems), such evaluation is common and acceptable. The main reason being the methods under this approach are less (if not minimally) reliance on data. Our method does fall under the category of the randomized algorithm, and for this purpose, we did evaluate each scenario 20x with different seeds for the random number generator and presented the results with its 95% confidence interval.
>
>  Moreover, pur presented scenario settings are specifically defined to be more difficult/challenging than randomized, especially for the covered and hidden case. Randomized cases-often lead to simple occlusion relationships (easy to observe in one or multiple poses, similar to “loose” cases). Last but not least, we do recognize the importance of understanding the effects of the different components and parameters of the method on its overall performance. To this end, we have taken the initiative to introduce an ablation study that systematically spans an escalating range of object quantities, ranging from 2 to 10 at intervals of 2. The scenarios, as depicted in Figure 1 of the attached pdf file, comprise 20 trials each, all adhering to a consistent planning time limit of 60 seconds per step. The results of these trials are documented in Figure 2 of the attached pdf file, which we hope would provide a comprehensive statistical insight into the performance of our method. Our method indicates more benefit, if the object number is relatively large (blue box areas in Figure 2), because we reuse the useful belief tree and avoid branch-cutting caused by the newly detected objects, which is more common in scenarios with more objects.
>
> Q2. I am confused about how the fake object is used and what the intuition behind it is.
>
> $\textbf{Answer}$: Thanks for pointing this out. Please refer to the “Fake object” section of the global comments file for more details.
>
> $\textbf{Response to limitation}$:
>
> L1. Limitations are only very briefly discussed in the conclusion.
>
> $\textbf{Answer}$: Thank you for the constructive question. Please refer to the “Limitation” section of the global comments file for more details.

---

### Official Review · Reviewer_UoQp · 2023-07-07

**Soundness:** 3 good
**Presentation:** 2 fair
**Contribution:** 2 fair
**Rating:** 5
**Confidence:** 3

**Summary:**

This paper proposes a method for mobile robots to efficiently search for objects in complex environments. They use a Partially Observable Markov Decision Process (POMDP) formulation and a planning algorithm called Growing Partially Observable Monte-Carlo Planning (GPOMCP) to improve the robot's success rate and speed in finding target objects.

**Strengths:**

* A Monte-Carlo planning method to perform object search in the indoor environment based on the grid-type 3D representation.

* A systematic framework and experiments demonstrate the effectiveness of the whole sense-plan-act pipeline.

* A novel design for growing state space using a belief tree.

**Weaknesses:**

* The whole framework seems complex and required carefully mannual design of parameters (e.g. $R_{max}$ and $R_{min}$ ). It is unclear how the sim-to-real gap will impact these parameters.

* It seems the baseline methods are POMCP and a variance of the proposed approach with replaced modules. It would be good if the authors can discuss or compare some other related object searching methods.

**Questions:**

* L123: the object pose is usually regarded as 6D or SE(3) although using quaternion.

* A concern regarding using growing state space and tree search is that the problem complexity when the object number increase. What is the computational time for the planner with a varying number of objects?

* How does the resolution of the grid representation impact the performance?

**Limitations:**

The authors have discussed the limitations.

---

> ### Author Rebuttal · Authors · 2023-08-08
>
> $\textbf{Response about weakness}$:
>
> W1. The whole...
>
> $\textbf{Answer}$: Thank you for providing valuable comments. While some parameters require manual configuration, it's worth noting that the majority of these parameters do not significantly impact the final performance of the framework. Most of them are not sensitive to the final performance. As an example, the performance does not change too much, if orders of magnitude between $R_{max}$ and $R_{min}$ satisfy $R_{max}>>R_{min}$. For instance, varying $R_{min}$ from -1 to -20 does not exert a substantial influence on performance outcomes. To elucidate this phenomenon, we present a comparison between the results using $R_{min}=-1$ (the case in our paper) and $R_{min}=-20$ for the $Hidden_1$ case in Table 1 of the attached pdf file.
>
> Few parameters may affect the performance a lot. For example, before successfully declaring, we need to update the belief of the log-odds of 8 grid values and then complete the declaring action by comparing the mean value of the minimal $n_{odds} = 2$ log-odds with the thresholds $C_d^o$ for the obstacle object and $C_t^o$ for the target object. The smaller $n_{odds}$ means that this condition is hard to be satisfied and we need to observe more directions of the objects. As $n_{odds}$ decreases, the removing and declaration actions become more dependable, owing to the augmented array of diverse observations derived from different orientations. Our method retains more useful branches after receiving observations and it will show obvious advantages when the target belief is harder to reach. With this goal, we increase $n_{odds}$ to 4 and 6 and show the statistical results of the scenario with 6 objects (Fig. 1) in Table~2 with 20 trials setting.
>
> In our opinion, the primary challenge of deploying our system to real-world robots lies in addressing errors and failures in the perception and navigation parts rather than our focused planning part, which constitutes our main contribution. If the considered parameters work fine in simulation and the perception and navigation methods work well, they should be fine to apply in real-world cases directly. This relatively direct sim-to-real parameter “transfer” is common in the applications of non-learning-based POMDP solvers to physical robots.
>
> W2. It seems...
>
> $\textbf{Answer}$: We appreciate the suggestion to consider more baselines for comparison. Please refer to the “Baseline methods” section of the global comments.
>
> $\textbf{Response to questions}$:
>
> Q1. L123: the...
>
> $\textbf{Answer}$: Sorry and thanks for pointing it out. 7D in our original formulation means 7 dimensions instead of degree of freedom. We have revised it into a 6D (Degree of Freedom) pose.
>
> Q2. A concern...
>
> $\textbf{Answer}$: Thanks for your insightful comments. Indeed, the computational complexity of our framework will mainly increase with the number of updating objects. These objects play a crucial role in both transition and visual observation functions, causing fewer sampled particles. Consequently, the computational expenditure associated with these declared objects becomes relatively economical, as it no longer necessitates exhaustive testing of occlusion relationships among all other objects. The good news is that commonly the number of the updating objects is limited by the declaring action. To facilitate a clearer understanding of how our method performs under varying object counts, we have conducted an ablation study. This study includes a comprehensive set of experiments, encompassing an increasing number of objects ranging from 2 to 10, with increments of 2. The scenarios, as depicted in Fig.1 of the attached pdf file, comprise 20 trials each, all adhering to a consistent planning time limit of 60 seconds per step. The results are meticulously documented in Fig.2. Our method will perform obviously better, if the object number is relatively large (blue box areas), because we reuse the useful belief tree and avoid branch-cutting caused by the newly detected objects, which is more common in scenarios with more objects.
>
> Q3. How does...
>
> $\textbf{Answer}$: This question presents a fascinating insight. The grid world's finer resolution offers a unique advantage in enhancing the reliability of probability updating within the grid. This advantage becomes particularly pronounced when dealing with smaller target objects. It's important to acknowledge that opting for this finer resolution does entail a slightly higher computational complexity (odds updating is very cheap for large datasets). Upon conducting tests, it becomes evident that the good grid resolution should be smaller than the minimum dimensions of the target object along both x and y axes. We are here to correct a typo in our previous version. All the experiments are implemented in 2 cm resolution instead of 5 cm. For comparison, we present statistical results for the $Covered_1$ scenario using 10 cm resolution: 24803.5±3792.3|39.9±6.9|60%. The result is poorer than the POMCP method without using fake object in success rate. To dissect this phenomenon, we manually executed a designated sequence of actions: “move_head_5-move_base_0-move_base_1-move_base_2-move_base_3-move_lift_2-move_base_0-move_base_1-move_base_2-move_base_3-remove_object_4-move_head_8-move_base_0-move_base_1-move_base_2-move_base_3” for our method with 2 cm and 10 cm resolution, and the resulting changes in probability distributions across the grid world are visually depicted in Fig.3. The salient observation is that when the resolution is relatively large, grid probability updates lose precision, particularly when the target object is situated close to an obstacle. The use of a larger resolution not only fails to enhance our method's performance but also introduces potential distortions in the information of the fake object. In short, we recommend the used resolution had better be smaller than the object sizes to improve the accuracy of updating odds values.

---

> > ### Comment · Reviewer_UoQp · 2023-08-20
> >
> > I appreciate the author's response and will maintain my ratings.

---

> > > ### Author Response · Authors · 2023-08-21
> > > **Appreciation for Your Responses**
> > >
> > > Thank you for giving us the previous comments about the ablation study. They are valuable.

---

> ### Comment · Area_Chair_JJFX · 2023-08-19
>
> Thank you for your review.
>
> The authors have provided a detailed response to your review. Please be sure to read it and reply indicating the extent to which the authors have addressed your initial questions and concerns.
>
> Best,\
> AC

---

### Official Review · Reviewer_uTWr · 2023-07-10

**Soundness:** 4 excellent
**Presentation:** 3 good
**Contribution:** 3 good
**Rating:** 7
**Confidence:** 4

**Summary:**

The paper presents a full object search pipeline based on a MCTS solver for POMDPs. The object are represented by a octo-grid containing the odds of the object being is the target or an obstacle. A 3D point cloud map of the environment *without any object* is given to the algorithm, allowing for easier object detection and segmentation. The authors propose a more efficient MCTS algorithm for their problem to allow tree re-use when a new state variable is added (i.e. a new object is detected). They test their method in gazebo environments using a simulated fetch robot, and their method performs better than POMCP with nearly always 100% success rate.

**Strengths:**

This is a very strong paper, systems-wise. The amount of work to create a full object search system such as this, even if just tested in simulation, is significant. The scientific contributions, for me, are 1: the minor (but still valid and significant) modification to POMCP to allow for a growing state-space. 2: the POMDP modeling of the object search problem, which was a very nice textbook-like example of how to model a challenging problem as a POMDP and actually solve it.

**Weaknesses:**

The are not many other baselines. There exists many other POMDP solvers than POMCP, and probably other non-MCTS approaches to object search.

I had trouble understanding the terminology "fake object", it's just the belief about the target object, which can be wildly inaccurate at first of course.

The existence of the point cloud map without any object "is assumed to be available before planning, which is a reasonable and
216 realistic assumption achieved by mapping the environment prior to planning.". I disagree, it is not a realistic assumption to have an object-free map prior to doing the task. Imagine deploying this is a home, a cluttered home at that. You would need to clear the whole house before mapping it! Please discuss alternatives for future work.

**Questions:**

What limitations do you think would pop-up when deploying in the real-world?

You touch on this subject, but how would you handle unsuccessful execution of primitives like move or lift? These will happen in the real-world.

**Limitations:**

There's no dedicated limitations paragraph or section, and it's hard to grasp from the paper. I think the object-free pre-existing map is a significant limitation, but I'd like the authors to do the mental exercise of writing down what would need to be done for this to be deployed in a home on a real-robot, and infer the limitations of their approach.

---

> ### Author Rebuttal · Authors · 2023-08-08
>
> $\textbf{Response about weakness}$:
>
> W1. The are not many other baselines......
>
> $\textbf{Answer}$: Thank you for your valuable comments. We appreciate the suggestion to consider more baselines for comparison. Please refer to the “Baseline methods” section of the global comments file for more details.
>
> W2. I had trouble understanding the terminology......
>
> $\textbf{Answer}$: Thanks for pointing this out. Please refer to the “Fake object” section of the global comments file for more details.
>
> W3. The existence of the point cloud map without any object......
>
> $\textbf{Answer}$: Thank you for pointing out the reasonability of the assumption. We partially agree with this point. A more complicated and realistic scenario is that the map of the environment is missing, and the navigation and perception errors will make the object search problem even more challenging. However, even with the given maps, the object search within a partially observable environment is still very challenging, which is our considered problem. In this paper, we focus more on the planning component of the system, while the perception is based on existing methods. If we consider the other planning-focused object search papers, like these papers ([1], [2]), a classical assumption is to offer the given maps and even the given object information, like object size, position, and orientation. The point cloud of the surrounding environment can be corrected once and our method can repeatedly use the map, which is not very expensive commonly.
>
> Of course, it is better to have stronger navigation in a full-unknown environment. In the future, we will apply the online semantic SLAM methods [3] to build the navigation map online, identify the workspaces, and complete the task without manual settings, like the autonomous continuous action domain instead of the discrete action domain. We have revised the statement about the reasonability of the assumption and state it in future work.
>
> Revise “In our framework, the point cloud map ${M}$ of the robot environment with some furniture and known objects is assumed to be available before planning, which is a reasonable and realistic assumption achieved by mapping the environment prior to planning.” into “To simplify the scenario and reduce the navigation effort, we assume the availability of a pre-built point cloud map (${M}$) of the robot's environment, including furniture and known objects, which can be created beforehand through environment mapping and reused during planning.”
>
> Add "Our future directions involve reducing manual settings, such as discrete pose candidates, dropping prior information, such as pre-built point cloud map  $\mathcal{M}$, and exploring continuous action domains."
>
> [1] A. Wandzel, Y. Oh, M. Fishman, N. Kumar, L. L. S. Wong and S. Tellex, "Multi-Object Search using Object-Oriented POMDPs".
> [2] Zheng, K., Sung, Y., Konidaris, G. and Tellex, S., 2021, September. Multi-resolution POMDP planning for multi-object search.
> [3] Rosinol, Antoni, Andrew Violette, Marcus Abate, Nathan Hughes, Yun Chang, Jingnan Shi, Arjun Gupta, and Luca Carlone. "Kimera: From SLAM to spatial perception with 3D dynamic scene graphs."
>
> $\textbf{Response to questions}$:
>
> Q1. What limitations...
>
> $\textbf{Answer}$: Thank you for the constructive question. Please refer to the “Limitation” section of the global comments file for more details.
>
> Q2. You touch on this subject...
>
> $\textbf{Answer}$: This is a very good question for further improving the practice of the whole framework. Unsuccessful execution may cause totally different challenges to our framework. Some of them are fine to solve based on our current framework. For example, the robot may fail to pick up a detected object and instead push it down, which will cause the data association to stop updating the standing object and introduce a newly detected object based on our framework. This may increase the number of steps in finding the target objects, but commonly fine in completing the final tasks. However, some failures in primitive actions may affect the final performance. There are several ideas about dealing with poor primitives. (1) Based on prior experiments, we can identify the failure rate of different actions in the given scenario and consider the obtained state transition probability in the POMDP transition function, which means considering the failure in planning and reducing their effect; (2) Another way is to improve the successful rate of the primitives using the following possible ways:
>
> Action Execution Monitoring: Implement a robust action execution monitoring system that can detect and handle failures during primitive action execution. This could involve incorporating sensors or feedback mechanisms to validate the outcome of each action. If a failure is detected, the system can trigger a reattempt of the action or take corrective measures to ensure successful execution.
>
> Better Action Implementation Methods: Enhance action implementation capabilities using more robust and advanced methods to mitigate the impact of uncertainties on primitive execution. Like the move_base and move_it toolboxes may not be robust enough in the complicated environment based on the default setting. For example, we are testing the locomotor, which is an extensible path planning coordination engine that replaces move_base and we are adjusting the used configurations in move_it to increase the removing successful rate.
>
> Robust Perception and Sensing: Enhance perception and sensing capabilities to mitigate the impact of uncertainties on primitive execution. By using more robust and diverse sensory inputs, the system can better adapt to changes in the environment and make informed decisions even in the presence of imperfect or incomplete information.
>
> $\textbf{Response to Limitation}$:
>
> $\textbf{Answer}$: Thanks for your comments. Please refer to W3 and the “Limitation” section of the global comments file for more details.

---

> > ### Comment · Reviewer_uTWr · 2023-08-20
> > **Further comment about map assumption**
> >
> > Thank you for the detailed reply which addresses most of my comments.
> > I thought about the map assumption some more, and the rephrasing saying "the map includes *known* objects" rather than being devoid of objects makes sense to me. Thank you. I think my concerns have been addressed and would like to reiterate that I feel pretty positive about this paper.

---

> > > ### Author Response · Authors · 2023-08-21
> > > **Appreciation for Your Responses and Positive View**
> > >
> > > Thank you very much for your valuable comments. Your input has highlighted an important research direction, particularly in relation to eliminating prior information from the map. Moving forward, we are committed to further developing this area and conducting additional real-world robot tests to enhance the practicality and robustness of our proposed framework.

---

> ### Comment · Area_Chair_JJFX · 2023-08-19
>
> Thank you for your review.
>
> The authors have provided a detailed response to your review. I realize that your review is favorable, but please be sure to read their response and reply indicating the extent to which the authors have addressed your initial questions and concerns.
>
> Best,\
> AC

---

### Author Rebuttal · Authors · 2023-08-08

$\textbf{Baseline methods}$

$\textbf{Answer}$: We appreciate the suggestion to consider more baselines for comparison. However, as a system-level application, it is generally fair because many object search approaches are based on the pure POMCP method with different problem settings, eg.[1] [2]. Our GPOMDP solver is designed specifically for problems with growing state space using belief tree reuse and a modified upper confidence bound for improved upper bound estimation. This method can be easily applied to other online POMDP solvers that use particle representations with belief trees, such as ABT[3] and DESPOT[4], offering a competitive advantage when dealing with problems involving a growing state space. Other approaches, which are mainly learning-based methods, eg.[5], often require substantial training datasets and may achieve a general performance level of 40%-60% success rate, which is lower than our results.

[1] Zheng, Kaiyu, et al. "A System for Generalized 3D Multi-Object Search."
[2] Wandzel, Arthur, et al. "Multi-object search using object-oriented POMDPs"
[3] Kurniawati, Hanna, et al. "An online POMDP solver for uncertainty planning in dynamic environment."
[4] Somani, Adhiraj, et al. "DESPOT: Online POMDP planning with regularization."
[5] Druon, Raphael, et al. "Visual object search by learning spatial context."

$\textbf{Fake object}$

$\textbf{Answer}$: Our fake object is essentially a state variable that represents a “guess” on the target object. Our fake object is not a real object or a future detected object. The real target object will be included in one of the later sub-state-vector $s_{o_1}$ to $s_{o_n}$. In our problem, initially, we do not know the objects in the environment, neither the number of objects nor their positions, which means the whereabouts of the target object are also initially unknown. To capture uncertainty about the target object compactly, we propose to “guess” the target object. It can be considered as an additional visible object, related to the target object, in transition and visual observation functions of the POMDP framework, but not detectable in the real world. It uses the same structure as the structure used to represent an object in the system, which includes an 8-cell occupancy grid map where each cell maintains a probability of whether the cell intersects with the target object. Under the POMDP framework, a belief over the fake object variable represents multiple guesses with different probabilities about the target object. Initially, the uncertainty about the target object can be quite large, but over time the belief in the position of the fake object tends to converge to a probability mass whose mode is at the true target object. The distribution of the fake object (grid world) saves the belief about the target object. But, you know, in POMDP problems, we need to predict the future probability of all girds in MCTS by going through the transition functions and observation functions, which will be time-consuming and memory-consuming. The application of the fake object becomes meaningful to involve the useful information (represent grid world), is very computational (fast for transition functions and observation functions) and memory (a low-dimension vector) cheap in exploration and rollout, and is friendly for coding (use the same transition and visual observation function as the other detected real objects). We have added some sentences for better explanation in the revision. Last but not least, we apologize for the seemingly odd “fake object” terminology and will revise the name to “guessed target object”.

The way to use the fake object is shown in Fig.4 of the attached file. Its grid world is updated based on the real-world observation from the camera and its position will be sampled and the other terms of the fake object will be re-initialized to use in the transition function and observation function of the Monte Carlo tree search. Then, this iterative process persists until the task is completed. We will add more explanations in the supplementary material.

$\textbf{Limitation}$

$\textbf{Answer}$: When deploying the system in the physical world, our main limitations mainly stem from the errors and failures from perception and navigation components, rather than our focused planning part. Our planning method should be fine to apply in real-world cases directly with reliable support from perception and navigation methods. Under the POMDP framework, for primitive actions, some of the errors and failures can be handled. However, large failures with multiple knock-on effects would be difficult to handle. An example is when the robot’s failed grasping resulted in the object falling under the table or falling and causing many other objects to fall too. Another limitation is that our framework is based on point cloud segmentation to identify different objects. It may generate incorrect bounding boxes for objects with extensive contact areas. This misinterpretation can result in erroneous data association, impacting belief updates and removing actions. Furthermore, YOLO and SIFT object detection techniques may struggle in scenarios with limited SIFT features and YOLOv5 toolbox coverage, especially in low-light environments. However, we believe that addressing these limitations is possible by incorporating more advanced perception and navigation methods.

In terms of the algorithm itself, scalability to problems with too many unknown objects in the environment can be difficult. In such scenarios, one could cluster objects hierarchically, and apply the proposed method to clusters of objects, rather than single objects, in the higher level of the hierarchy. Application at the single object level can then be limited to a medium-sized area after the system has a better understanding of the target object.

We have added a new paragraph to emphasize the limitations of our method in the revised paper by summarizing the above points.

---

### Decision · Program_Chairs · 2023-09-21

**Decision:**

Accept (poster)

**Comment:**

The paper considers the problem of searching for a target object in a potentially cluttered environment using a mobile robot that is subject to a limited field-of-view, localization errors, and occlusions caused by objects in the environment. Given access to a 3D point cloud and occupancy grid of the environment, the paper formulates the search problem as a partially observable Markov decision process (POMDP) and proposes a Monte Carlo tree search (MCTS)-based solver (GPOMCP) that is able to reuse the previous belief tree under certain conditions when a new state (e.g., a newly observed object) is added. The system is evaluated on four simulated mobile manipulation environments and shown to outperform a POMCP baseline.

The paper was evaluated by four reviewers who largely agree on the paper's core strengths and weaknesses. Among them, the reviewers emphasize the significance of the object search system that the paper proposes---this is a strong systems paper that clearly describes a non-trivial framework and its implementation that addresses an important problem in robotics. Beyond the value of the systems-level integration and the merits of the underlying POMDP formulation, they also appreciate the modifications to the POMCP required to account for a growing state space. However, the reviewers are also in agreement about the weaknesses of the paper as originally submitted. Key among them is the lack of appropriate baselines both with regards to quantitative comparisons as part of the experimental evaluation as well as qualitative comparisons to related work that was not included in the original submission. The authors address this, stating in part that several existing search methods are based on the POMCP method to which they compare, while other methods require substantial training. The paper should be updated to better justify the choice of baselines and include a qualitative comparison to the methods that Reviewer 2WUu references. Additionally, several reviewers point out that the paper lacks an explicit section that identifies the limitations of the method or really any discussion of its limitations. The authors provide a detailed discussion of the limitations in their response and state that this discussion will be added to the paper. The reviewers and AC appreciate the authors' efforts to address the reviewers' initial questions and concerns, which make this a solid systems-level paper.